# Passive Attention in Artificial Neural Networks Predicts Human Visual Selectivity

**Thomas A. Langlois**[1,a,b], **H. Charles Zhao**[1,a], **Erin Grant**[c], **Ishita Dasgupta**[d],
**Thomas L. Griffiths**[2,a,e], and **Nori Jacoby**[2,b]

[1]T.A.L. and H.C.Z. contributed equally to this work.
[2]T.L.G. and N.J. contributed equally to this work.
[a]Department of Computer Science, Princeton University
[b]Computational Auditory Perception Research Group, Max Planck Institute for Empirical Aesthetics
[c]Department of Electrical Engineering and Computer Sciences, UC Berkeley
[d]DeepMind, New York
[e]Department of Psychology, Princeton University

## Abstract

Developments in machine learning interpretability techniques over the past decade have provided new tools to observe the image regions that are most informative for classification and localization in artificial neural networks (ANNs). Are the same regions similarly informative to human observers? Using data from 79 new experiments and 7,810 participants, we show that passive attention techniques reveal a significant overlap with human visual selectivity estimates derived from 6 distinct behavioral tasks including visual discrimination, spatial localization, recognizability, free-viewing, cued-object search, and saliency search fixations. We find that input visualizations derived from relatively simple ANN architectures probed using guided backpropagation methods are the best predictors of a shared component in the joint variability of the human measures. We validate these correlational results with causal manipulations using recognition experiments. We show that images masked with ANN attention maps were easier for humans to classify than control masks in a speeded recognition experiment. Similarly, we find that recognition performance in the same ANN models was likewise influenced by masking input images using human visual selectivity maps. This work contributes a new approach to evaluating the biological and psychological validity of leading ANNs as models of human vision: by examining their similarities and differences in terms of their visual selectivity to the information contained in images.

## 1 Introduction

The last decade has witnessed the rise of artificial neural networks (ANNs) that can match and even exceed human performance on a variety of perceptual and cognitive tasks, ranging from image recognition [1] to natural language processing and reinforcement learning [2]. Alongside the rapid development of these technologies, a significant body of work aimed at improving the interpretability of these systems and comparing them to biological ones has also grown [3, 4, 5, 6, 7]. In computer vision, techniques for probing which visual regions ANNs "attend to" when classifying images have been developed to visualize the receptive fields of convolutional layers as well as regions of a visual input that most influence the class activations of the models [8, 9, 10, 11, 12]. In neuroscience, researchers began to quantify the functional fidelity of leading ANNs as models of the human visual system using both neural and behavioral benchmarks [13, 14, 15]. Finally, cognitive scientists have developed techniques to compare the structure of ANN learned representations to human

psychological representations [16, 17, 18]. All these efforts have contributed to our understanding of the biological and psychological validity of leading ANNs as models of biological vision, beyond just assessing their performance on standard object categorization benchmarks.

Many previous analyses of the correspondence between ANNs and human vision have focused on the *representations* used by the systems. However, a natural question is whether ANNs *select* information in the same way, and in particular whether they attend to the same visual regions as humans when extracting information for visual object recognition and localization. Prior work has developed ANNs trained explicitly to predict human visual gaze [19], and even incorporated simulated foveated systems into the model design [20]. In addition, work comparing human attention to computational attention [21, 22, 23, 6, 7] revealed that computational attention tends not to resemble human attention. However, this work also provides evidence that using human attention to supervise the training of network architectures devised to emulate the parallel pathways in the human visual system can improve performance, and yields more human-like visual features [6]. Still, relatively little work has attempted a more comprehensive examination of how a wide range of ANNs compare to multiple human behavioral measures, using the large variety of interpretability techniques that are now available to probe what visual information ANNs use.

Methods for gaining insight into what is learned by ANNs started with "passive" attention gradient-based approaches designed to reveal which visual inputs influence the class activation score the most [24]. More advanced techniques using deconvolution and guided backpropagation methods followed [9, 25] as well as techniques that introduced novel design alterations, such as global average pooling layers and class activation mapping (CAM) to localize class-specific visual regions in the input images [10]. Finally, more general approaches that could be applied to architectures without global average pooling [26] appeared, with some of the most recent contributions proposing class activation mapping techniques that do not rely on gradients at all [27]. Aside from this range of "passive" techniques, computer scientists have also developed CNN models that incorporate end-to-end trainable attention modules [28, 29] as a means for both improving interpretability and boosting performance. The full range of techniques now available for visualizing the information that is most relevant to ANNs offer an unprecedented opportunity to compare their results to biological analogues such as estimates of human attention, discrimination accuracy, and visual recognition over image regions.

Since the early 19th century [30] vision scientists devoted to the study of biological vision have also developed a variety of experimental techniques for estimating the visual information used by the primate visual system when engaged in similar perceptual and cognitive tasks such as visual search, localization, and recognition. Among these are measures of visual change sensitivity (discrimination accuracy), visuospatial memory (spatial localization estimation), as well as explicit reports of visual recognizability. In this work, we obtained estimates from six different perceptual tasks, including explicit visual recognizability estimates using a recent behavioral task [31, 32] (Fig. 1A and Fig. 2A), a two-alternative forced choice (2AFC) change sensitivity task (Fig. 1B and Fig. 2A), eye-tracking fixations (Fig. 1C and Fig. 2A), and a spatial localization task (Fig. 1D and Fig. 2A).

We then compared them to estimates of ANN visual selectivity using a variety of pretrained models and visualization techniques including guided backpropagation and techniques based on class activation mapping. We used a range of model types, including early convolutional networks like AlexNet [1], as well as recent state-of the-art models, such as Visual Transformer (ViT) [33] and EfficientNet [34] models.

We find that only a select class of ANN models and passive attention techniques capture the shared variance across all human visual selectivity measures. This work contributes to current efforts aimed at evaluating the biological and psychological validity of contemporary ANNs by investigating the similarity between artificial and biological vision systems at the level of the visual inputs rather than the learned representations or correspondence to patterns of neural activation in the visual cortex [13]. We see our primary contributions as two-fold: (1) In a departure from prior work, we use a larger range of human behavioral measures, ANN models, and attention techniques in our comparisons. These reveal a significant range in the degree to which different models combined with different attention techniques produce human-like results. In addition, they show that more human-like ANN attention captures a shared component of the joint variability between the different human measures, rather than any particular human measure. (2) We provide causal evidence that more human-like ANN attention directly influences both human classification performance as well as predictions made by ANNs.

## 2   Human visual selectivity measures

We computed 6 behavioral measures with 25 images, performing experiments in which a total of 4,050 participants took part (see Table 1 in the Appendix). We ran experiments for 3 distinct human behavioral tasks for each image (see Fig. 1 and Fig. 2 for representative examples). A total of 1,575 participants took part in the discrimination accuracy experiments (an average of 63 participants for each of the 25 images), and a total of 225 participants took part in each of the image patch ratings tasks (9 participants per image). Finally, the spatial memory serial reproduction chain experiments were completed by a total of 2,250 participants (an average of 90 participants for each of the images). Participants were recruited anonymously over Amazon Mechanical Turk (AMT), and provided informed consent. Participants were paid approximately $7 per hour. Details of the experimental procedures, design, and map estimation for each task can be found in Appendix Fig. S1.

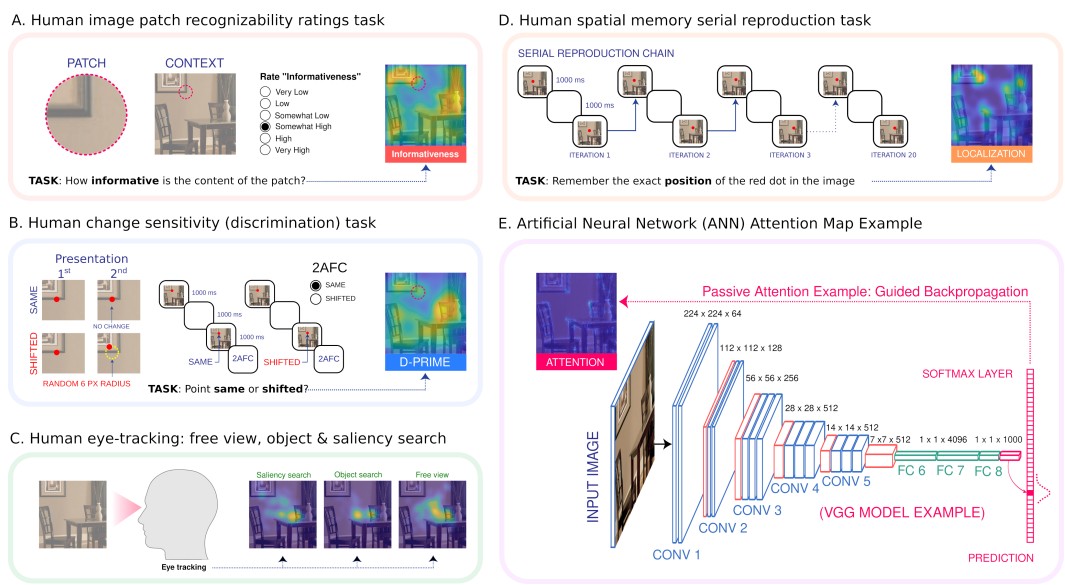

Figure 1: Human behavioral tasks, and ANN attention. A. Informativeness patch ratings task. B. Discrimination accuracy 2AFC task. C. Eye-tracking for free search, object search, and saliency search. D. Spatial memory serial reproduction task. E. ANN attention (passive attention example). Details of the experimental procedures and map estimation are included in Appendix Fig. 1.

1. **Visual recognizability.** We adopted a recent behavioral task [31, 32] designed to measure the informativeness and recognizability of local image regions by using explicit self-reports (see Fig. 1A). In the task, participants view small circular image patches sampled from full images and rate how "recognizable" or "informative" the content of the patch is on a six-point Likert scale ranging from a rating of "Very low recognizability" to "Very high recognizability". The patches were sampled from a regular 12 x 12 grid over the image. Fig. 2A shows representative examples of the results following averaging over all the ratings in different spatial areas of the images, smoothing, and interpolation to produce continuous maps for each image. We ran 25 experiments with an average of 9 participants per image (see Appendix for details of the map generation procedure). A total of 225 participants took part in the patch ratings experiments. Participants were paid $2 to complete 144 experimental trials.

2. **Change sensitivity (Discrimination).** We measured change sensitivity using a two-alternative forced choice (2AFC) discrimination task. In this task, participants viewed an image with a small red dot superimposed on it for 1000 milliseconds. Following a 1000 millisecond delay, the same image was presented again with the dot in either the same exact location or in a slightly displaced location (see Fig 1B). Participants were then asked to indicate if the dot was shifted or unchanged in the second presentation. Crucially, the locations of the initial dot locations were sampled densely from all possible locations on a regular grid that spanned the dimensions of the image, in order to measure changes in visual change sensitivity over the entire image (see Appendix for details). This task measures changes in visual acuity conditioned on different visual areas in an image, and has

been used as a proxy for measuring variable encoding precision of different image regions [35]. We ran 25 experiments (one for each image) with an average of 63 participants per experiment. The overall number of participants for the discrimination tasks equalled 1,575. Participants were paid $1.50 to complete 120 trials in the discrimination task.

3. **Visuospatial localization.** We used a recent behavioral paradigm based on serial reproduction that can reveal intricate spatial memory priors that guide visual localization estimation in humans [35]. In this paradigm the first participant views a point superimposed on an image and then reproduces its location from memory. The next participant views the same image but with the point located in the position reconstructed by the previous participant. As in the "telephone game," the process is repeated, forming a chain of participants. For each image, there were a total of 20 iterations in the chains, for 250 initial random seed dot positions. This experimental procedure is known to reveal the spatial landmarks in visual scenes that bias human allocentric visuospatial representations (see [35] and Fig. 1D). We ran 25 experiments (one for each image) with an average of 90 participants each. The overall number of participants was 2,250. Participants were paid a base rate of $1.00 for completing 105 trials in the spatial memory experiment but could earn up to $1.50 depending on accuracy in the task. Additional details are provided in the Appendix.

4. **Fixations.** Finally, we used an existing dataset of human fixations obtained via eye-tracking when human participants were engaged in a free-viewing task, a cued object search task, and a saliency search task [36] (Fig. 1C). We used the published data from the 75 experiments reported in [36].

## 3    ANN models, passive and active attention

We evaluated three standard deep convolutional neural network architectures (AlexNet [1], VGGNet [24], ResNet [37]), as well as two state-of-the-art architectures (Vision Transformer (ViT) [33] and EfficientNet [34]). For each of these models, which do not have active attention modules, we obtained attention maps using a range of passive methods described below. We also evaluated two built-in end-to-end trainable attention modules, described below. All models were pretrained on ImageNet 2012 [38], CIFAR-100 [39], or Places365-Standard [40]. Unlike the ImageNet and CIFAR-100 datasets, which are comprised of images of objects, Places365-Standard is comprised of images of complex natural scenes. See Table 2 in the Appendix for a list of all models.

### 3.1    Passive attention

We used gradient-based techniques including guided backpropagation methods, as well as more recent techniques based on class activation mapping (see Fig. 1E for a schematic example of passive attention). The methods based on guided backpropagation effectively try to compute the sensitivity of the model's output with respect to each pixel in the input image, using various techniques for increasing the signal in these maps and decreasing noise. These methods include guided backpropagation (GBP) [25], guided gradients times the image (GBPxIM) [41], and SmoothGrad with guided backpropagation (SGBP) [42]. The methods based on class activation mapping compute linear combinations of the activation maps in the final convolutional layer of the model in order to determine the discriminative regions of the image used by the model. These methods include Grad-CAM [26], Score-CAM [27], and CAMERAS [43]. See the Appendix for details on these passive attention methods.

### 3.2    Active attention

We evaluated two different active attention modules. These active attention modules are trainable, and they learn to generate masks which are applied to the input image (or to intermediate convolutional layers). These masks are effectively explicit attention maps, so we do not have to use passive attention methods to try to discern the models' attention. These attention modules can be incorporated into essentially any standard CNN architecture, so we chose a few for which we were able to obtain pretrained weights (see the Appendix for details).

One attention module is described in the paper Learn to Pay Attention [28], which we will refer to as LTPA. LTPA inserts attention at three intermediate convolutional layers within a VGGNet architecture. The other active attention module is used in Attention Branch Networks (ABNs) [29].

ABN has a separate "attention branch" that runs in parallel with a "perception branch," and is based on class activation mapping [10].

## 4 Passive attention predicts shared variance across human measures

**Human experiments.** Fig. 2A shows representative results from each of the human behavioral experiments. We found that the human maps were correlated with one another though the correlations varied from $r = 0.14$ to $0.86$ (see Fig. 2B) due to variations in the visual regions that were the most implicated from one task to another. We obtained a single factor that captures the maximal amount of shared variance across all behavioral maps ("Human PC"; see Fig. 2C) by computing a linear combination of the results of the six experiments via Principal Component Analysis (see SI Appendix for details and formal description). The Human PC was correlated with each of the six behavioral measures ($r = .75, .74 , .42, .81, .81$ and $.73$ for the informativeness maps, change sensitivity maps, spatial localization maps, and each of the fixation maps, respectively).

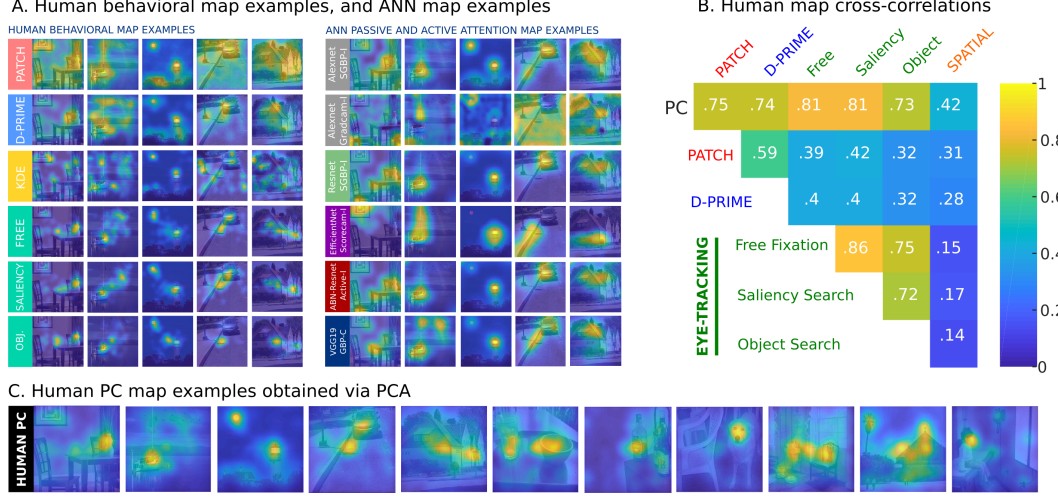

Figure 2: Human behavioral task maps and ANN maps. A. Representative examples of the human maps obtained for the (1) patch "informativeness" ratings task (2) The discrimination accuracy task (3) the serial reproduction spatial memory task (4) Free fixations (5) saliency search fixations and (6) cued object search fixations. Examples of ANN maps for the same images are also shown. B. Cross correlations of all the maps, including the linear combination of all the maps that predicts the maximal shared variance (Human PC), are shown. While most methods are relatively highly intercorrelated, there are clear differences. Fixations were the most highly intercorrelated ($r = .72$-$.86$), while spatial memory Kernel Density Estimates (KDEs) are only weakly correlated ($r = .14$-$.17$) to the fixations, in line with previous findings [35]. Factor loadings of each of the six measures to the Human PC were uniformly high ($r = .42$-$.81$). C. Human PC map examples.

**ANN maps.** We then compared the maps computed by the ANN attention methods (Fig. 3A; Raw ANN maps are included in Appendix Fig. S3). We found that ANN maps varied significantly in terms of their level of "smoothness" depending on the attention method. Because of this, and in order to compare the human and ANN maps in a way that is agnostic to the raw smoothness of the ANN maps, we introduced a smoothing parameter when comparing the two. We optimized the smoothing parameter for each of the ANN maps based on the correlation of the result to each of the human behavioral maps including the Human PC. We did this by applying the same smoothing to each of the individual 25 image ANN maps, and then computing the average Pearson correlation of those maps to the corresponding human maps for each of the 25 images. We repeated this process for each ANN attention method, and for each human task.

**The relation between ANN and human maps.** Next, we explored the relation between each of the human measures (including the Human PC) and the ANN attention maps. Fig. 3A shows the optimal correlations between human and ANN maps for each human map type including the Human PC. The peak correlations between the Human PC maps and the most highly correlated ANN maps exceed the

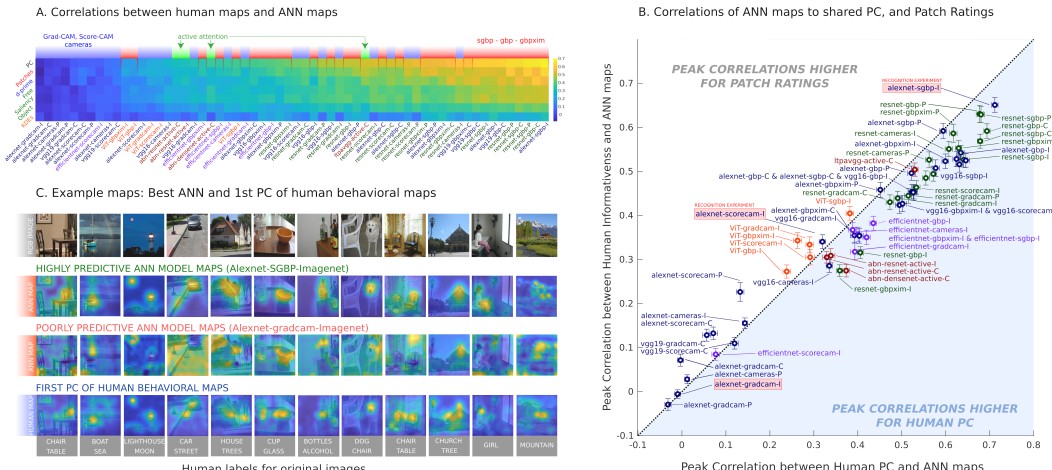

Figure 3: Human behavioral maps and ANN attention. A. Cross-correlations between each of the human maps (including the shared Human PC) and the ANN maps. Results are sorted according to the average peak correlations across the six human measures and PC maps. Blocks highlighted in red correspond to ANN maps obtained using SGBP, GBP, or GBPxIM methods. Blocks highlighted in green correspond to maps obtained from LTPA and ABN active attention models. Blocks with blue highlights correspond to maps obtained using CAM-based methods. Results show that maps obtained using GBP-based attention methods are better aligned with human behavioral maps than both CAM-based results and active attention results ($p < 0.001$, and $p < 0.001$). The naming convention for passive attention maps is <architecture>-<passive attention method>-<I/C/P for ImageNet/CIFAR-100/Places365>. The naming convention for active attention maps is <attention module>-<architecture>-active-<I/C for ImageNet/CIFAR-100>. B. Correlations between all ANN maps and Human PC (x-axis) and the correlation between all ANN maps and the Human Informativeness ratings. Error bars were estimated from 100 bootstrapped samples of the human data. The red boxes indicate the ANN maps used for the human speeded recognition task. The blue shaded area represents the scenario where the Human PC correlations to ANN maps (optimized to Human PC maps) are higher than the corresponding correlations between human informativeness maps and ANN maps (optimized to human informativeness maps). C. Representative examples of the ANN maps that were highly predictive of the Human PC maps ($r = 0.71$, $p < 0.001$), and not highly predictive of the Human PC maps ($r = 0$) are shown. Also shown are the Human PC maps for the same images.

peak correlations of the same ANN maps to each of the six behavioral maps (Fig. 3B; PC average $r = 0.71$, compared with $r = 0.36$-$0.65$ for each of the human maps). This suggests that the peak ANN maps predict the shared component of the variance between all the human measures rather than the variance of any particular human map type (such as human fixations or discrimination accuracy for instance). This is significant because it indicates that some intrinsic aspects of visual information contained in the images captured by the peak ANN models are predictive of human visual selectivity regardless of the behavioral task, and in spite of the systematic differences between them [35]. Fig. 3B illustrates this fact for the human patch ratings task results. For most ANN maps, the correlation values under the diagonal in the plot indicate that the peak correlations of the ANN models to the Human PC were significantly higher than the peak correlations of the same ANN models to the Human patch ratings results. Fig. S4 illustrates the same finding for all individual human behavioral measures. In addition, Fig. 3B shows the top correlations for all ANN maps reaching a peak of $r = 0.71$ for the SGBP method applied to the AlexNet network pretrained on ImageNet (alexnet-SGBP-I). In a separate analysis, we repeated the smoothing parameter fitting using split-half cross-validation, and found that performance of smoothed hold-out test set maps using smoothing parameters fit to random training set maps produced nearly identical ranges in peak correlations to the human PC (between $r = 0.73$ and $r = -0.01$ for the training set, and between $r = 0.72$ and $r = -0.04$ for the testing set), as well as a nearly identical rank order in peak correlations to the human PC. Details of this analysis are included in the SI Appendix (see Fig. S8).

A natural question concerns whether the peak correlations achieved by the leading ANN maps are due to the model architecture, attention method, or training set. Fig. 3A shows correlations of the

ANN maps to the human measures by attention type. We used the average correlations across the 6 measures and images as a dependent variable and found that the attention method category explains 35.9% of the variance, while architecture and training set categories explain 25.3% and 1.5% of the variance, respectively (see Fig. 3A for the average performance across all map types, and for each of the human map types). This result suggests that attention type plays the most significant role in our findings. Surprisingly, SGBP applied to one of the simplest architectures (the AlexNet network pretrained on ImageNet) showed the highest peak correlation to the Human PC ($r = 0.71$), and was significantly more predictive of the Human PC and human patch ratings than most of the other ANN maps ($p < 0.001$ with Bonferroni corrections applied).

Passive attention models were consistently predictive of the human maps across behavioral tasks (See Appendix Fig. S2, S3 and S4). However, there was significant variation in the performance of these methods. For the human PC results, Guided-backpropagation based attention results (GBP, SGBP and GBPxIM; averaged $r = 0.51$) consistently outperformed active attention results (averaged $r = 0.38$, $p = 0.001$), and these in turn outperformed Class Activation Mapping (CAM) based attention results (Grad-CAM, Score-CAM and CAMERAS, averaged $r = 0.27$, $p = 0.001$). In addition, we found that the resnet models (averaged $r = 0.54$) produced more human-like maps than the remaining models (averaged $r$ ranged between 0.25 - 0.42, $p < 0.001$ in all cases; with the Bonferroni correction applied). Finally, comparing results from the model types for which we could extract both active and passive attention maps reveals that guided-backpropagation based attention produces more human-like results than active attention architectures (See SI Appendix for details).

## 5 Validation experiment: ANN visual selectivity influences human recognition

We showed that the shared component of the variability across human behavioral measures is best predicted by attention maps computed using a particular class of passive attention methods, suggesting that recognition performance in both humans and machines is derived from the same visual information in images. However, this finding is based on indirect correlational evidence. In order to provide direct evidence for this claim, we tested whether human recognition performance is improved in real-time using the leading ANN maps via a speeded recognition experiment. We reasoned that recognition performance in humans will be better for images masked using their own attention maps ("correct masking") obtained from the best ANN maps (as defined by their peak correlation to the Human PC) compared with images masked using maps obtained for different images ("incorrect masking," see Fig. 4A for the masking procedure, and Fig. 4B for the speeded recognition task design). Furthermore, we predicted that the difference in recognition performance on this task between these two conditions (correct masking vs. incorrect masking) would be greater if the masks are generated using the leading ANN maps than if they are generated using ANN maps that are very dissimilar to the Human PC maps.

To test these hypotheses, we ran three additional behavioral experiments with a total of 3,600 participants recruited from AMT. Each participant viewed briefly flashed masked images (for 200 milliseconds), and had to select the best descriptors from a set of word pairs obtained from a separate labelling experiment in which a total of 160 participants took part (see Appendix). Masked images were generated by an element-wise product of a grayscale image with the map produced by the ANN model, see Fig. 4A. *Correct masking* consisted in masking an image with the map produced by the ANN for that image. *Incorrect masking* consisted in masking an image with the attention map produced by the ANN for different images. Representative examples of the masks along with the original color images are shown in Fig. 4C.

Based on the rankings of the ANN maps in terms of their peak correlations to the Human PC, we predicted that human recognition accuracy should be significantly more sensitive to the visual regions revealed by SGBP applied to the AlexNet model than those revealed by the peak maps produced by the same model using the scorecam or Grad-CAM attention methods. Paired t-tests revealed a significantly larger difference ($p < 0.001$) between the $d'$ scores across map types in the correct masking condition (blue bars in Fig. 4D) relative to the differences in the $d'$ scores across map types in the incorrect masking condition (red bars in Fig. 4D; see Appendix for definition of $d'$ scores). This interaction confirms the prediction that human recognition accuracy is in fact significantly *more* sensitive to the visual regions revealed by one of the models and attention methods with the highest peak correlations to the Human PC, but less sensitive to those revealed by passive attention techniques applied to a model that yielded significantly lower peak correlations to the Human PC, even though

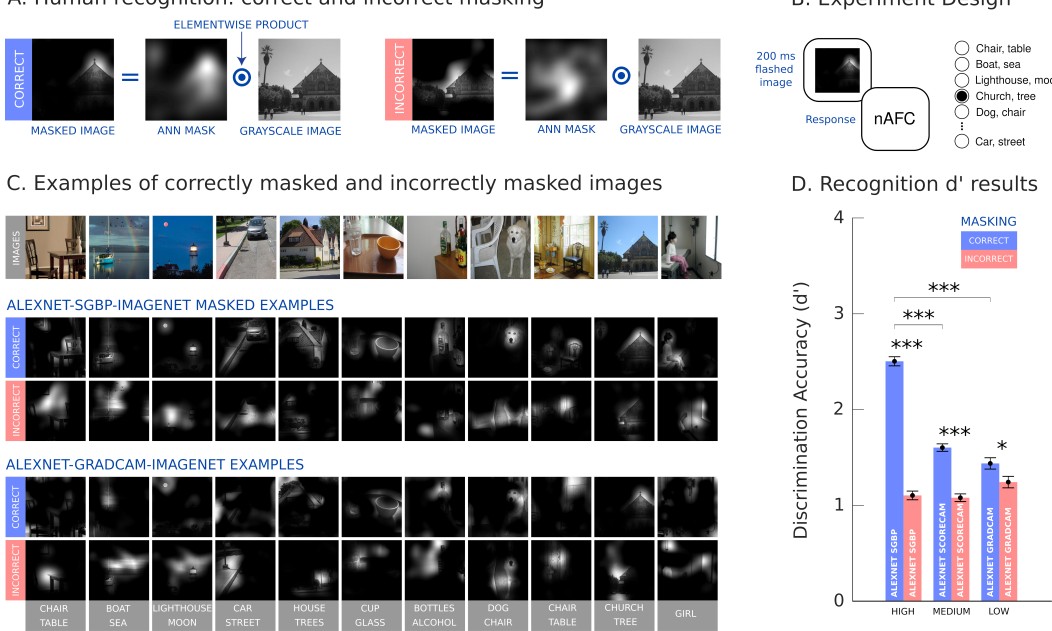

Figure 4: Human recognition results. A. Masking procedure. We masked images by an elementwise product of the grayscale image with the ANN map. B. Experimental design. Participants viewed a masked image for 200 milliseconds. They then selected the word descriptors that best described the image. C. Example original images, and corresponding masked versions obtained from the ImageNet-pretrained alexnet-SGBP maps (high peak correlation to Human PC), and from the same model also pretrained on ImageNet using Grad-CAM attention (low peak correlation to Human PC). D. Overall $d'$ results for the experiments using SGBP, Score-CAM, and Grad-CAM maps. Results reveal significantly higher $d'$ for correctly masked images (blue bar) relative to incorrectly masked images (red bar) for all 3 ANN map types ($p < 0.001$, $p < 0.001$, $p < 0.05$ for SGBP, Score-CAM, and Grad-CAM maps, respectively). In the correct masking condition (blue bars), there was a significant difference between $d'$ results for the SGBP maps relative to the $d'$ results for the Score-CAM and Grad-CAM maps ($p < 0.001$). Error bars were computed from 1000 bootstrapped samples of the human responses.

correct masking did produce a boost in recognition accuracy over incorrect masking for all three map types ($p < 0.001$, Fig 4D). $d'$ scores broken down by individual images can be found in the SI Appendix, including the same results shown in terms of simple accuracy (% correct) rather than $d'$ (See SI Fig. S6).

## 6    Validation experiment: human visual selectivity influences ANN recognition

The results of the human speeded recognition experiments suggested that we evaluate the same prediction in the opposite direction by asking whether ANN classification performance is similarly sensitive to the visual regions revealed by the human behavioral maps. To do this, we masked images using the six different human behavioral maps including the Human PC maps (Fig. 3). As with the human recognition experiments, we evaluated if ANNs show better performance in classifying correctly masked images rather than incorrectly masked images (*i.e.,* images paired with the mask from a different image). Furthermore, we tested whether ANNs show improvements when the correct masks were obtained from the behavioral measures that had the highest peak correlations to the ANN maps (Fig. 3). See Fig. 5A for an illustration of the whole procedure.

**Measuring recognition.** In this experiment, we evaluated how masking affects ANN recognition directly. We therefore used the ANN's classification of the original, unmasked image as a baseline, and then compared it to classification on the corresponding (correctly or incorrectly) masked image.

Similar classifications in both cases would indicate good performance. We defined similarity across classifications as follows. For a given ANN, we took the top-1 category (*i.e.,* the category predicted with highest confidence) for the unmasked image, and computed its rank in the predictions on the masked image ($r$). We then divided this rank by the total number of categories ($N$) to normalize for differences in the number of categories across ImageNet, CIFAR-100, and Places365 trained models. This new quantity, ($r/N$), inversely tracks recognition quality—it is lower when recognition is good, and higher when recognition is poor. To convert it into a measure that gives *higher* values for better recognition and *lower* values for poorer recognition (as with the human $d'$ accuracy metric), we use $N/(r + N)$ as our final measure. We refer to this measure as the *inverse-rank* and computed it for all (correctly and incorrectly) masked images. Note that due to domain shift we expected the inverse rank to be less than one even in the correctly masked case.

**Results.** We computed the inverse-rank across all models, for all types of human maps (Section 2) as well as for both kinds of masking (correct, incorrect). We found that the correctly masked images are universally more recognizable (have higher mean inverse-rank) than incorrectly masked images, across each of the different human maps (Fig. 5B, $p < 0.001$). This finding validates our core prediction that ANNs should be sensitive to visual regions revealed by the human measures.

Masked images were also more or less easy to recognize based on the human map type, and we found a significant main effect of behavioral map type on the overall mean inverse-rank ($F(6,176) = 49.65$, $p < 0.001$) in the predicted direction: human maps that had higher peak correlations to the ANN maps also produced higher overall inverse-rank scores. We also found a significant interaction between masking condition (correct vs. incorrect) and map type, indicating that different human behavioral maps have a direct impact on recognizability (the *difference* in inverse-rank between the correct and incorrect masking conditions ($F(6,176) = 6.54$, $p < 0.001$). This finding is key, because it confirms a change in recognition performance that is predicted from the differences in overall peak correlations of the ANN maps to the different human maps. Like the interaction we observed with the human recognition experiment, it shows that behavioral measures that had higher peak correlations to the ANN maps (such as the patch rating maps) also gave higher inverse-rank score differences across masking conditions, while others, like the fixation maps, gave smaller inverse-rank score differences across masking conditions. More details of the analysis and results can be found in Appendix Fig. S6.

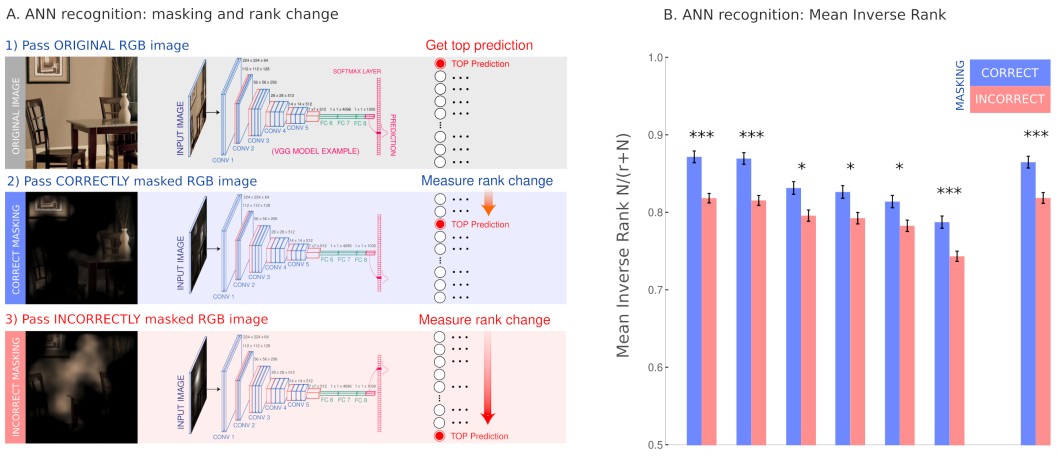

Figure 5: ANN recognition results. A. Measuring recognition. As outlined in the text, we directly evaluate how masks derived from human maps affect ANN recognition by examining how the rank of the top image category prediction when classifying an unmasked image changes when that same image is masked with either the correct or incorrect mask. B. Mean Inverse-rank across different human maps grouped by correct vs incorrect masking, averaged across all ANN models and images. All human maps give a higher inverse-rank for correct vs incorrect masking, validating the main hypothesis that ANNs are sensitive to visual regions highlighted by human maps. We also find an interaction in the predicted direction: the effect of correct masking tended to be greater when maps were from the behavioral results that were the most highly correlated to the ANN maps overall.

# 7 Discussion

Using a range of human behavioral measures, ANN models and attention techniques (Fig. 2, Fig. 3), we attempted a comprehensive examination of the similarities and differences between humans and machines with respect to their visual selectivity to image information. In a departure from prior work, we found that ANN maps are optimally predictive of a latent human visual selectivity feature that captures the joint variability between our human behavioral measures (Human PC; see Fig. 3).

Surprisingly, simple architectures and passive guided backpropagation based attention techniques showed the highest peak correlations to the human data, and performed significantly better than the maps produced by active attention or class activation mapping based techniques. These results suggest that the same visual regions are informative to humans and machines. In addition, comparing models for which we could extract both active and passive attention maps revealed that active attention architectures consistently produce less human-like results (SI appendix section D.2). This is noteworthy because active attention models have often been conceived with the goal of emulating human attention mechanisms.

In another departure from prior work comparing human attention to ANN attention, we validated our correlational findings by running two causal experiments. In the first, we took the ANN maps and used them to mask images presented to human participants. We found that humans were better at classifying images that were masked with the correct ANN map compared with incorrect maps, and that the difference in performance between these two conditions was greater when the ANN maps used were more highly correlated to the Human PC maps (Fig. 4). In the second experiment, we used human maps to mask images and measured the effect on recognition performance for the ANN models. Again, we found that incorrect masking was more destructive to ANN recognition performance than correct masking (Fig. 1). We also found that the change in performance between the two conditions tended to be greater when the masking was done using human maps that were the most highly correlated to the ANN maps.

Our results suggest that the regions that are discovered by attention techniques in both humans and ANNs are indeed mutually important for recognition. Moreover, these results show that we can use off-the-shelf ML methods (i.e., ANN and interpretability methods) to produce saliency maps that can predict human visual selectivity just as well as custom models (i.e. non-deep-learning visual-attention models) designed specifically for that purpose. We include predictions made by two such models: Graph-based Visual Saliency (GBVS) and Boolean Map based Saliency (BMS) [44, 45] in the SI Appendix (see Fig. S2 and S4).

One of the main findings of this paper is the fact that more human-like ANN maps tend to be more predictive of the shared component of the joint variability between human measures rather than any single human measure. Although this finding may be due in part to a reduction in measurement noise from summing the individual behavioral maps, the fact that we observed systematic variations in the visual regions that are implicated depending on the behavioral task (Fig. 3B), as well as relatively high internal reliability for each of the estimates (split-half correlations between $\rho = 0.63 - 0.92$, see SI Appendix), suggests otherwise.

The main limitation of this work is the inclusion of a relatively small number of images. This is due to the large number of participants needed in order to create detailed estimates for all the behavioral maps for each of the images (requiring over a hundred participants for every image). In addition, further work will be required to fully explain the factors that contribute to greater similarity between artificial attention and human visual selectivity, and ways in which we can devise systems that are better models of biological vision. In addition, while making artificial networks more human-like has practical advantages for improving their interpretability, pitfalls include introducing undesirable biases [46, 47, 48]).

Overall, our results pave the way for developing new psychologically relevant benchmarks for evaluating leading ANN models, beyond comparing them to the neural basis of biological vision, or the distributions of their learned representations to the structure of human psychological representations. These results showcase new ways of combining the perspectives of machine learning and cognitive science towards developing more human-like intelligent systems.

**Acknowledgements**

This work was funded by Princeton University (Grant number 1718550 from the National Science Foundation) as well as the Max Planck Society.

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
