**Appendix to "Passive Attention in Artificial Neural Networks Predicts Human Visual Selectivity"**

## A  ANN models, passive attention, and active attention

In the paper we used the following naming convention. The naming convention for passive attention maps is <architecture>-<passive attention method>-<I/C/P for ImageNet/CIFAR-100/Places365>. The naming convention for active attention maps is <attention module>-<architecture>-active-<I/C for ImageNet/CIFAR-100>.

Table 1: Human experiments

| Experiment | Map short name | Approx. participants per experiment | Total participants for all experiments[1] | New experiment |
|---|---|---|---|---|
| Patch ratings | PATCH | 9 | 225 | Yes |
| Discrimination accuracy | D-PRIME | 63 | 1,575 | Yes |
| Spatial memory | SPATIAL | 90 | 2,250 | Yes |
| Free-viewing fixations | FREE | (22) | (22) | No [1][2] |
| Saliency search fixations | SALIENCY | (20) | (20) | No [1][2] |
| Object search fixations | OBJECT | (19) | (38) | No [1][2] |
| Recognition validation | N/A | 1,200 | 3,600 | Yes |
| Labelling | N/A | 5 | 160[3] | Yes |

Table 2: Evaluated deep learning models

| Model | # params | Model ref. | Impl. ref. |
|---|---|---|---|
| CIFAR AlexNet | 2.50M | [2] | [3] (MIT) |
| CIFAR VGG-19 (w/ BatchNorm) | 20.09M | [4] | [3] (MIT) |
| CIFAR ResNet-110 | 1.73M | [5] | [3] (MIT) |
| ImageNet AlexNet | 61.10M | [2] | [6] (BSD-3) |
| ImageNet VGG-16 (w/ BatchNorm) | 138.37M | [4] | [6] (BSD-3) |
| ImageNet ResNet-101 | 44.55M | [5] | [6] (BSD-3) |
| ImageNet EfficientNet-B0 | 5.29M | [7] | [8] (Apache 2.0) |
| ImageNet ViT-S/16 | 22.05M | [9, 10] | [8] (Apache 2.0) |
| Places AlexNet | 58.50M | [2] | [11] (CC BY) |
| Places ResNet-50 | 24.26M | [5] | [11] (CC BY) |
| CIFAR LTPA VGG | 19.99M | [12, 4] | [13] (GPLv3) |
| CIFAR ABN ResNet-110 | 3.06M | [14, 5] | [15] (MIT) |
| CIFAR ABN DenseNet-BC ($L = 100, k = 12$) | 1.12M | [14, 16] | [15] (MIT) |
| ImageNet ABN ResNet-101 | 62.58M | [14, 5] | [15] (MIT) |

1. **ANN models.** We evaluated several different CNN models, some with active attention modules and some without. Some of these models were trained on CIFAR-100 [17], meaning that they take in 32x32 images as input and classify into 100 possible classes; some were trained on ImageNet 2012 [18], meaning that they take in 224x224 images as input and classify into 1000 possible classes; and some were trained on Places365-Standard [11], meaning that they take in 224x224 images as input and classify into 365 possible classes. The particular models we evaluated are listed in Table 2. The

---

[1]For our experiments we are counting the number of AMT Human Intelligence Tasks (HITs) that were completed. We did not exclude AMT workers from completing multiple HITs. For the eye movement data, numbers in parentheses represent unique human participants.

[2]Parentheses indicate publicly available data, see [1] for details. Images and eye fixation data were obtained from https://data.mendeley.com/datasets/8rj98pp6km/1.

[3]For technical reasons, we only retained the data for 125 participants out of the full 160 that were recruited for the labelling experiment.

models in the top part of the table were evaluated with passive attention methods, and the models in the bottom part of the table have active attention modules.

2. **Passive attention methods.** Passive attention methods have been developed to provide insights into which parts of an input image a model is attending to. In addition to early gradient-based techniques [19, 20], we used class activation mapping techniques [21, 22]. We adapted open-source PyTorch implementations of these methods [23][4] (released under the MIT License). For each method, we computed attention maps with respect to the top class prediction made by the model. For Grad-CAM, Score-CAM, and CAMERAS, activation maps were derived from the last convolutional layer. We describe each of the methods below:

(a) Guided backpropagation (GBP) [20] – This computes an imputed version of the gradient. It is the same as standard backpropagation except it prevents the backward flow of negative gradients by zeroing them out. By doing so, it uses higher layers to "guide" backpropagation to lower layers.

(b) Guided gradient $\times$ image (GBPxIM) [24] – This is the same as the output of guided backpropagation except it is multiplied by the (normalized) input image. Roughly, this gives a first-order approximation of the effect of setting any given input pixel to 0.

(c) SmoothGrad with guided backpropagation (SGBP) [25] – This method attempts to address the noisiness in raw gradient visualizations. The authors posit that this noisiness is because the gradient may fluctuate sharply at small scales, which seems plausible especially given that, due to ReLU activation functions, the output generally is not even continuously differentiable. To address this, SmoothGrad adds Gaussian noise to the original input and performs guided backpropagation, repeats this to generate a sample of sensitivity maps, and then averages these together to produce the final sensitivity map. We used a sample size of 30, and we set $\sigma$ such that the noise level is 10%.

The next three methods are based on class activation maps (CAMs) [26]. The original CAM method relies on a global average pooling (GAP) layer between the final convolutional layer and the fully-connected layer in an image classification CNN. A GAP layer simply computes the average value over each activation map in the final convolutional layer, and uses these averages as inputs into the fully-connected layer. The CAM is then defined as the linear combination of these final activation maps, weighted by the target class's weights in the fully-connected layer. This CAM indicates the discriminative regions of the image used by the CNN to identify that class. Since not all CNN architectures have a GAP layer, several closely related techniques have been developed that do not rely on a GAP layer. We describe the three methods we used below:

(d) Grad-CAM [21] – This stands for gradient-weighted class activation mapping. This is the same as standard CAM except that instead of relying on a GAP layer, Grad-CAM uses gradients to weigh each activation map.

(e) Score-CAM [22] – There are several issues with gradient-based approaches like Grad-CAM, such as gradients being noisy and tending to vanish. Therefore, Score-CAM avoids gradients altogether. Score-CAM is the same as standard CAM except that instead of relying on a GAP layer, Score-CAM uses forward-passing scores to weigh each activation map. An activation map's forward-passing score for a given class is defined as the model's score for that class when the input image is masked by the activation map.

(f) CAMERAS [27] – CAM methods generally rely on the low-resolution activation maps of the final convolutional layer, resulting in saliency maps that may be imprecise. CAMERAS addresses this by up-/down-sampling the input image to multiple resolutions (we used resolutions of 100x100, 224x224, and 1000x1000) and running each of these through the ANN. The resulting gradients and activations are averaged and then used to produce the final, fused class activation map.

We used each of the above passive attention methods to acquire attention maps from each of the models in the top part of Table 2.

---

[4]Note that we made some modifications to this code, and these modifications are present in our supplementary materials. In particular, we fixed the SmoothGrad implementation so that $\sigma$ is computed correctly, and we generalized the guided backpropagation implementation (and therefore all the other methods that rely on guided backpropagation) so that it works for architectures other than AlexNet and VGGNet. Our code is available at `https://github.com/czhao39/neurips-attention`.

3. **Active attention methods.** Relatively recently in the area of computer vision, some models have been developed which have active attention modules built into their architectures, so that these models explicitly attend to certain locations in the input. We evaluated two different active attention modules, which we describe below:

   (a) Learn to Pay Attention (LTPA) [12] – We refer to this architecture by the title of its paper, LTPA. In particular, we used the (VGG-att3)-concat-pc model defined in the paper, and apply attention before the max-pooling layers. The model is based off of a VGGNet architecture, with three attention estimators at intermediate layers within the CNN. Of the three attention estimators, we used the attention maps produced by the middle estimator in our analyses.

   (b) Attention Branch Network (ABN) [14] – ABN also uses an end-to-end trainable attention module. ABN consists of three modules: feature extractor, attention branch, and perception branch. The feature extractor and perception branch are constructed by splitting a baseline CNN into two parts. The attention branch is placed after the feature extractor and is based on class activation mapping. We used the resulting CAM as the attention map in our analyses.

We evaluated the LTPA and ABN models listed in the bottom part of Table 2.

## B    Non-ANN models of human attention

The focus of this work is to evaluate visual selectivity of ANNs in relation to human visual selectivity. In particular, we do *not* claim that ANN saliency represents the state-of-the-art in modeling humans. Nevertheless, it is still instructive to compare ANN saliency to non-ANN models of human attention (i.e., models designed specifically to predict human attention). We examined two state-of-the-art non-ANN models of human attention:

   1. Graph-Based Visual Saliency (GBVS) [28] – GBVS first computes standard biologically-inspired feature maps (Gabor filters, contrast maps, and luminance maps), then uses a graph-based approach to compute activation maps on these feature maps which are meant to highlight locally "unusual" regions, then uses a graph-based approach to normalize these activation maps in a way that concentrates the mass on these maps, and finally sums these together. This approach is naturally parallelizable, suggesting biological plausibility.

   2. Boolean Map based Saliency (BMS) [29] – As opposed to most models which focus on properties of local image patches (e.g., contrast and rarity), BMS tries to model a global perceptual phenomenon found to be relevant to human visual attention—figure-ground segregation. In particular, BMS computes a saliency map by detecting surrounded regions in an image.

## C    Human visual selectivity estimation

### C.1    Experimental design and procedure

**"Informativeness" patch ratings task.** We used the procedure described by [30, 31] to generate dense "meaning" maps for all 25 images (taken from the database of images used by [1] for which detailed eye-movement fixation patterns were available). For the patches, we extracted circular image regions from a 12 by 12 regular grid over the entire image. The patches were extracted from high-resolution versions of the images that were full-color 2430 by 2430 pixel images. The diameter of the patches was 442 pixels (see SI Appendix Fig. S1A). During the experiment, we presented each of the patches along with a small thumbnail of the full image that included a green circular marker over the image to indicate where the patch was extracted from, for context. Participants rated the "informativeness or recognizability" of the image content revealed by each of the patches using a Likert scale (1 = "Very low recognizability", 2 = "Low recognizability", 3 = "Somewhat low recognizability", 4 = "Somewhat high recognizability", 5 = "High recognizability", 6 = "Very high recognizability"), see SI Appendix Fig. S1A). In the experiment, the terms "informativeness" and "recognizability" were used interchangeably. Participants rated all 144 patches for a given image per experiment, and we obtained judgments from 9 unique participants for each image patch over AMT. Participants were paid $2 for their participation. The exact instructions at the start of the experiment were as follows: "In the task, you will see a circular image patch along with a thumbnail of the full image from which the patch was taken to provide context. A circle over the full thumbnail image

will indicate the location of the patch. your job is simply to rate the content revealed inside each circular image patch (NOT the full image) in terms of how RECOGNIZABLE or INFORMATIVE it is using a 6-point Likert scale ('very low', 'low', 'somewhat low', 'somewhat high', 'high', 'very high'). There is no right or wrong answer." SI Appendix Fig. S1D shows the final map results for this task, for all 25 images.

**Change sensitivity discrimination task.** We used the same design and procedure described in [32]. We started by producing a regular grid of possible point locations that spanned the full area of each of the images (see SI Appendix Fig. S1B). The grid points were 7 pixels apart. During the task, participants saw an image presented for 1000 ms with a red point displayed over it (SI Appendix Fig. S1B). Following a 1000 ms blank delay, the image reappeared with the point either in the same exact location relative to the image or in a shifted position. In the "shifted" condition, the point was shifted by 6 pixels somewhere along a circular radius around the original point location, sampled at random. The second display remained for 1000 ms on the screen and was followed by a 2AFC ("red dot same", or "red dot shifted"). Participants could take as long as they liked to choose a response, although they had to complete the experiment within one hour before the experiment expired on AMT. We obtained responses from a total of 9 participants for each grid point, and for each condition ("same" or "shifted"). The full instructions at the start of the experiment were as follows: "In this experiment, you will see two images presented one after the other. These images will have a red dot placed over them. Your task is to determine if the red dot is in the same spot relative to the image for both images in the pair, or if the red dot appears displaced the second time it is presented. NOTE: The displays will be shown in random positions on the screen, even in cases when the red dot is placed in the EXACT SAME spot over the image! So part of the challenge is to ignore the random shifting of the overall display, and focus on the RELATIVE positions of the dots in relation to the images, ignoring the random overall displacements." For the discrimination experiment, compensation was $1.5, and included 120 trials. Participants could take part in as many discrimination experiments as they wished.

**Spatial memory task.** We used the same design and procedure introduced by [32] to estimate spatial memory priors. Participants were presented with an image with a red point initialized somewhere over the image for 1000 ms. The location of the point was sampled from a uniform distribution. Participants were instructed to reproduce the exact location of the point relative to the image from memory as accurately as possible. Exact instructions were: "In this task, a background image display with a red dot over it will be shown for some time, and will be followed by a blank screen. The background image will reappear but without the red dot inside. Your task will be to place the dot in the exact position where you previously saw it (relative to the background)!". Overall positions of the displays, including the point and image, were shifted by a random horizontal and vertical offset between 0 and 80 pixels on the screen canvas (in order to avoid a strategy of marking the absolute positions of the points on the screen). The response of the participant was sent to another participant on AMT who performed the same task, with their response becoming the stimulus for the next participant (and so on). A total of twenty iterations (generations) of the process were completed for each chain (There was a total of 250 chains for each image, beginning as random point location initializations over the image). We ended each experiment after approximately 12 hours. Typical participation included 105 trials, and the average time needed to complete the task was about 12-14 minutes. A typical experiment included about 90 participants. Compensation for taking part in the task was dependent on performance (between $1.4 and $1.5). Participants could take part only once per experiment, but could take part in more than one experiment (for more than one image). We only retained the chains that reached the full twenty iterations. Participants completed 10 practice trials prior to moving on to the 95 experimental trials.

## C.2 Cross-participant reliability in the behavioral measures

We evaluated the internal reliability of our behavioral estimates using split-half reliability. We computed 100 random splits of the data for each human behavioral task and measured the average split-half correlations for the 100 split-half pairs with the Spearman-Brown correction. The results were: $\rho =.85$ for the patch ratings estimates, $\rho =.88$ for free fixations, $\rho =.92$ for cued object search fixations, $\rho =.87$ for saliency search fixations, $\rho =.63$ for the spatial memory KDEs, and $\rho =.75$ for the $d'$ discrimination accuracy maps.

## C.3   Map estimation

**"Informativeness" patch map estimation.** To compute full maps from the patch ratings, we averaged the ratings made for all patches at a given pixel location and ratings made at neighboring pixel locations weighted by an isotropic Gaussian kernel (as a weighted interpolation of the ratings at a given pixel with the ratings at neighboring pixel locations). To increase the dynamic range, we then exponentiated the resulting map by squaring each of the individual values in the map to obtain the final maps. SI Appendix Fig. S1D shows the final results for all the images.

**Change sensitivity map estimation.** We obtained $d'$ values for each of the discrimination grid points by using the 2AFC responses obtained for each grid point (see formula for computing $d'$ included in Appendix F.1). We then convolved the grid of raw $d'$ values with a fixed Gaussian kernel. Next, we generated full continuous $d'$ map estimates by interpolating between the grid points using cubic interpolation. Finally, to increase the dynamic range, we exponentiated the final result by squaring each of the map values. SI Appendix Fig. S1B shows the results including the raw $d'$ grid point values, the smoothed $d'$ grid point values before the interpolation, and the smoothed $d'$ interpolated maps (final discrimination accuracy maps) for one of the images. SI Appendix Fig. S1D shows the final interpolated change sensitivity estimates for all the images.

**Spatial memory Kernel Density Estimation (KDE).** We computed KDEs using the data from the last (20th) iteration of the serial reproduction chains. For each point in the 20th iteration, we computed a Gaussian kernel centered at that point with a diagonal covariance matrix. We then summed all of the Gaussian kernels and then normalized to get the final KDE. Final KDEs are shown in SI Appendix Fig. S1D.

**Free fixations, cued object search fixations, and saliency search fixations.** To compute maps of the fixations data, we used the raw eye-movements data in [1], which consisted in (x,y) coordinates of the fixations over the images, and used the same kernel density estimation procedure used for the spatial memory task (see above) to obtain the final maps. These are shown in SI Appendix Fig. S1D.

## D   Comparing human and ANN maps

### D.1   Principal Component Analysis (PCA) for estimating Human PC

In order to estimate the shared component of the variance between all the human behavioral maps, we completed a Principal Component Analysis (PCA) with the concatenated human maps as input (a 250,000 x 6 matrix containing 6 vectorized versions of the concatenations of the maps across all 25 100 x 100 images, for each map type). We z-scored (standardized) each of the 6 2,500 x 100 human map concatenations prior to vectorizing them, and then concatenated them into the full 250,000 x 6 matrix for the input to the PCA. We used MATLAB's *pca* function to obtain the 6 PCA coefficients (loadings) for each of the human map vectors that explain the maximal shared variance between them. We then obtained the Human PC as a linear combination of the 6 human map types using these factor loadings. Because we did not want to overrepresent the three human fixations maps (which were highly intercorrelated ($r$ = 0.72-0.86) we downweighted each by multiplying them by a factor of $\sqrt{\frac{1}{3}}$ to obtain the final PC map. The PC map images for all 25 images are shown in SI Appendix Fig. S1E. The first PC explains 53.3% of the variance in the human behavioral measures (whereas the next three components explain 19.9%, 12.4% and 6.8% of the variance).

### D.2   Predicting human PC and behavioral maps using ANN attention

For each human map type (including the human PC), we computed the peak average correlation between the 25 image maps (which were resized to 100 x 100 images) and the corresponding attention maps produced by each of the ANN methods. For each ANN method, we searched over a smoothing parameter range of $\sigma = 0 - 30$ and selected the one that produced the peak average correlation (over all 25 images) of the ANN method maps to the corresponding human maps by smoothing each using MATLAB's *imgaussfilt* function (see SI Appendix Fig. S2A). Barplots showing the ANN methods sorted by peak average correlation to each of the human behavioral maps are shown in SI Appendix Fig. S2B-H. Error bars for each of the ANN methods shown in each barplot were obtained by resampling the human data with replacement and estimating 100 new human maps, and then taking the standard deviation of the averaged correlations of each of the 100 bootstrapped map

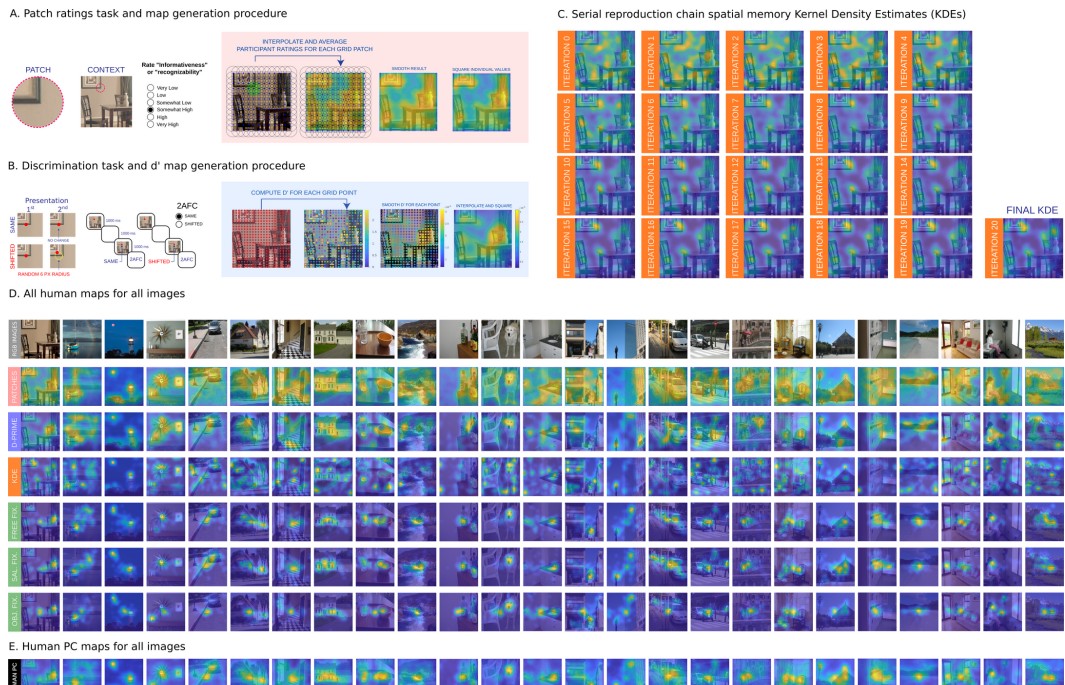

Figure S1: Human behavioral tasks, map generation procedures, and results for all images. A. "Informativeness" patch ratings task, and map estimation. For each pixel, maps were obtained by averaging the ratings given for patches that overlapped with that pixel, including ratings given for neighboring locations weighted by an isotropic Gaussian centered at the pixel. B. Discrimination task and map generation procedure. Participants completed 2AFC tasks for each grid point sampled from a regular grid of locations over the image. For each point, we computed $d'$ using the 2AFC responses, smoothed the grid $d'$ values, and interpolated between them to obtain continuous change sensitivity estimates over the entire images. For both experiments we exponentiated the final map values by squaring each value to increase the dynamic range. C. Spatial memory localization estimates and serial reproduction procedure. Participants were instructed to remember precise point locations from memory. Each response from a given iteration in the chain was forwarded to the next participant in the subsequent iteration. Kernel Density Estimates (KDEs) of the results at each of the chain iterations are shown, including the final (20th) estimate. D. Behavioral map results for all tasks, including the 3 fixations (see [1] for details). E. Human PC maps for all images.

estimates to the optimally smoothed ANN method maps across all 25 images. For the human PC, the 100 bootstrapped map estimates were obtained by a linear combination of the 100 bootstrapped sample map estimates obtained for each of the 6 behavioral maps using the weights (loadings) from the PCA (see section above for PCA details). Pairwise t-tests comparing the peak ANN method maps to all other ANN method maps revealed significantly higher correlations for the maps produced by AlexNet pretrained on ImageNet using SGBP relative to most of the remaining ANN methods for human maps except the object search fixation maps ($p < 0.001$; We applied the Bonferroni correction for multiple corrections; see SI Appendix Fig. S2). Qualitative examples of the ANN maps, including unsmoothed and smoothed examples (fit to the human PC maps), are shown in SI Appendix Fig. S3A. Examples shown include ANN maps that were among the top predictors of the human PC maps, as well as the lowest predictors of the human PC maps (SI Appendix Fig. S3A).

Overall average peak correlations achieved between the smoothed ANN maps and the human PC maps were higher than the peak correlations achieved between the ANN maps optimized to each individual human map. This finding is illustrated in Fig. F3B of the main text for the human patch ratings results, and for all 6 human behavioral maps in SI Appendix Fig. S4. This indicates that the ANN maps capture a shared component of the variability across all human behavioral measures rather than the unique variance captured by any one of the individual human maps.

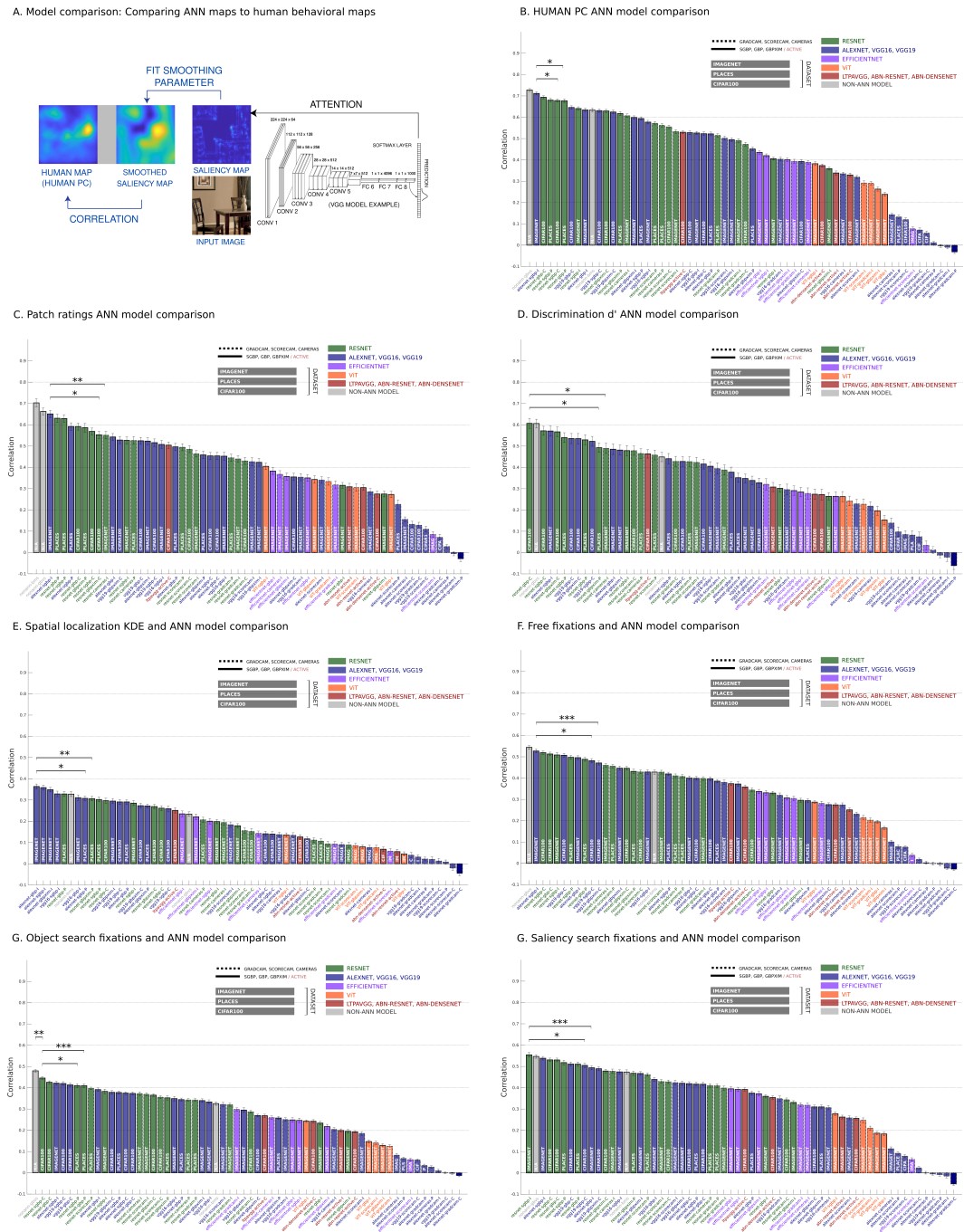

Figure S2: ANN attention and human visual selectivity. A. Schematic of the ANN and human map comparison procedure. We fit a smoothing parameter that maximized the average correlation (across all 25 images) of the smoothed ANN maps to the corresponding human maps. The schematic illustrates an example for a single image, correlating a smoothed ANN attention map with the corresponding human PC map. B. Sorted peak average correlations across all 25 images for all ANN maps to the human PC maps. The maps produced by ImageNet pretrained AlexNet probed using Smooth Guided Backpropagation (SGBP) were significantly more highly correlated to the human maps than most of the remaining ANN maps ($p < 0.001$; with the Bonferroni correction for multiple comparisons applied). C. results for the "informativeness" patch ratings maps. As with the Human PC results, the same ANN maps outperformed the remaining ANN maps ($p < 0.001$). Panels D-H show the results for all remaining human maps. They show similar results, with guided backpropagation methods producing ANN maps that were more predictive than others overall ($p < 0.001$).

A. Human PC maps and ANN maps (smoothed and unsmoothed)

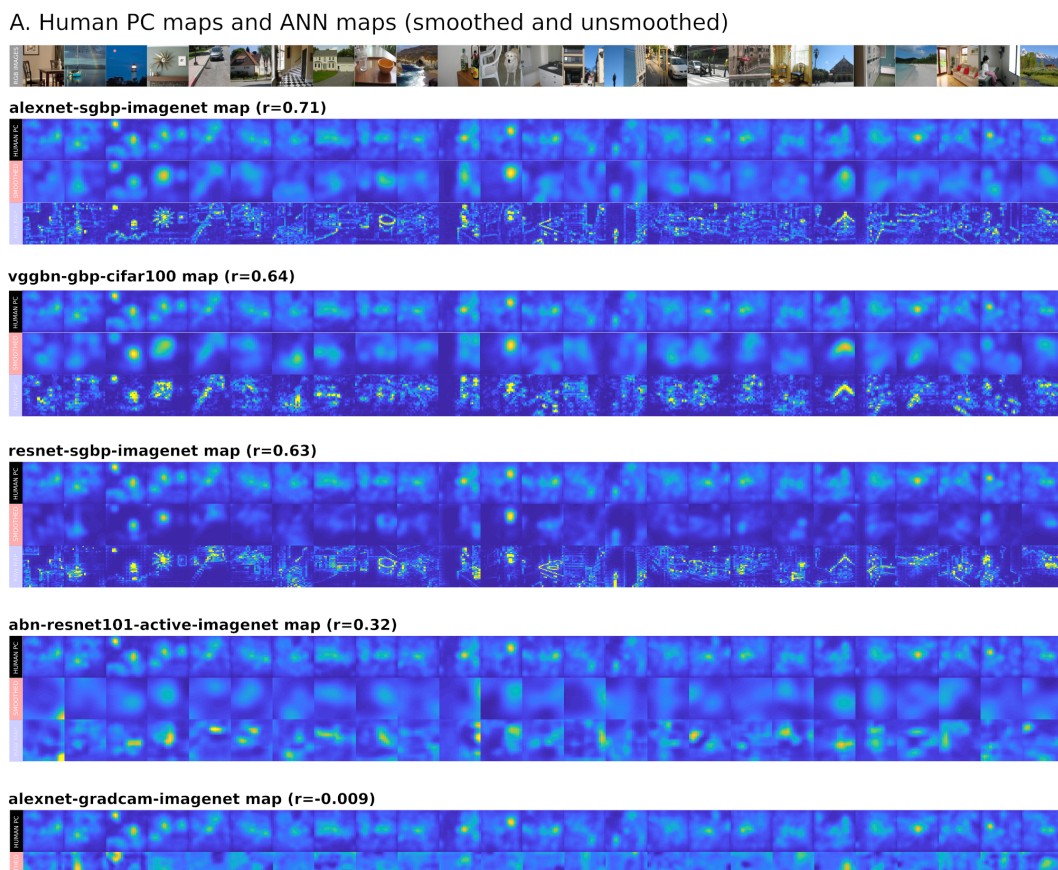

Figure S3: Human PC maps, and ANN maps (optimally smoothed and raw unsmoothed examples). A. Human PC maps (first row of each group of three). Second row in each group of three shows optimally smoothed ANN maps, and third row in each group shows the unsmoothed raw ANN maps, for all images. Examples for some of the best performing ANN maps are shown, including intermediate and worse performing examples. The peak average $r$ correlations to the human PC maps achieved with optimal smoothing are shown for each ANN map type.

With respect to passive and active attention, we observed more human-like results using passive attention techniques applied to the same models for which we could extract active attention maps. In particular, we found that LTPA (which is a VGGNet model with an added active attention module, and is trained on CIFAR-100) achieved a peak correlation with the human PC of $r = 0.53$, while SGBP applied to VGG19 (also trained on CIFAR-100) achieved a higher peak correlation of $r = 0.64$ ($p < 0.001$). Similarly, ABN ResNet-110 (which is a ResNet-110 model with an added active attention module trained on CIFAR-100) achieved a peak correlation of $r = 0.31$, while SGBP applied to "vanilla" ResNet-110 produced a much higher peak correlation of $r = 0.68$ ($p < 0.001$). Finally, ABN ResNet-101 (trained on ImageNet) achieved a peak correlation of $r = 0.32$, while SGBP applied to ResNet-101 (also trained on ImageNet) has a higher peak correlation of $r = 0.63$ ($p < 0.001$). These results indicate that comparing identical discriminator networks for which we could obtain both passive and active attention maps shows that active attention networks produce far less human-like results.

## D.3 Statistical comparison of averaged correlations

When comparing averaged correlations between different methods or training sets, we computed the averaged peak correlations for each condition and for each image. For each comparison, we

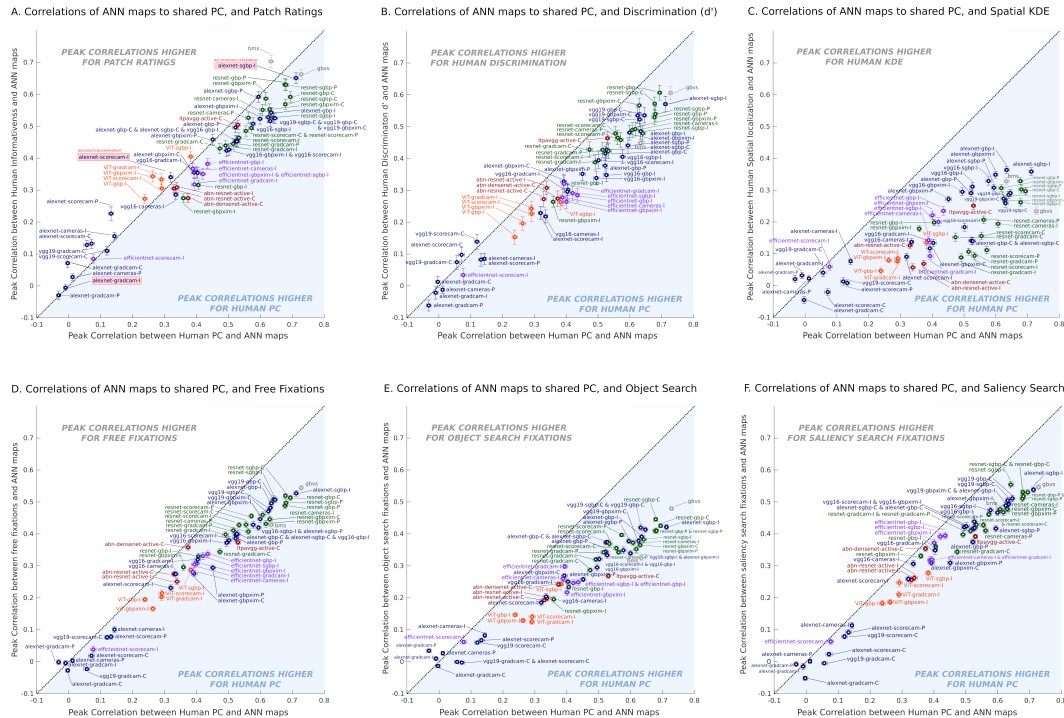

Figure S4: Peak correlations between all ANN maps and Human PC (x-axis) and peak correlation between all ANN maps and each of the human behavioral maps (y-axis). Error bars were estimated from 100 bootstrapped samples of the human data. A. Results for the patch ratings experiment shown in Fig. F3B in the main text. Points plotted below the dotted diagonal line inside the blue shaded area correspond to ANN map types that achieved peak correlations to the Human PC maps that were higher than the corresponding peak correlations of the same ANN maps to the particular human map types shown. Results for discrimination accuracy $d'$ maps, spatial memory KDE maps, and the three eye-movement fixation maps are shown in panels B-F. The fact that peak average correlations achieved by the higher-performing ANN maps to the Human PC maps are overall higher than those achieved by the same ANN maps optimized to each individual human map indicates that the most human-like ANN maps capture a shared component of the variability across all human behavioral measures rather than the unique variance captured by any one of the individual behavioral maps. Note, we also include peak correlations achieved by non-deep learning computational attention methods in each subplot. These are the peak correlations for Graph-based Visual Saliency (GBVS) and Boolean Map based Saliency (BMS) methods. These results show that we can use off-the-shelf ANN and interpretability methods to produce saliency maps that can predict human visual selectivity just as well as custom models (i.e. non-deep-learning visual-attention models) designed specifically for that purpose.

then performed a paired t-test across all 25 images for the two averaged vectors. When multiple comparison were reported, we used the Bonferroni correction.

# E Human recognition experiments: design

## E.1 Human recognition experiment: experimental design

Participants completed a speeded categorization task for 10 images (trials). The 10 trials were comprised of a unique permutation of a multiset of 5 correctly masked image cases, and 5 incorrectly masked image cases. The 10 images were a set of unique images sampled without replacement from the full set of 25 images. Each set did not contain more than one image containing the same object label (labels were obtained through crowdsourcing, see below). For example, a set could only contain one car and street image. We generated 200 sets of 10 images with 5 correctly masked cases and 5 incorrectly masked cases each, ensuring that the total number of correctly masked and incorrectly

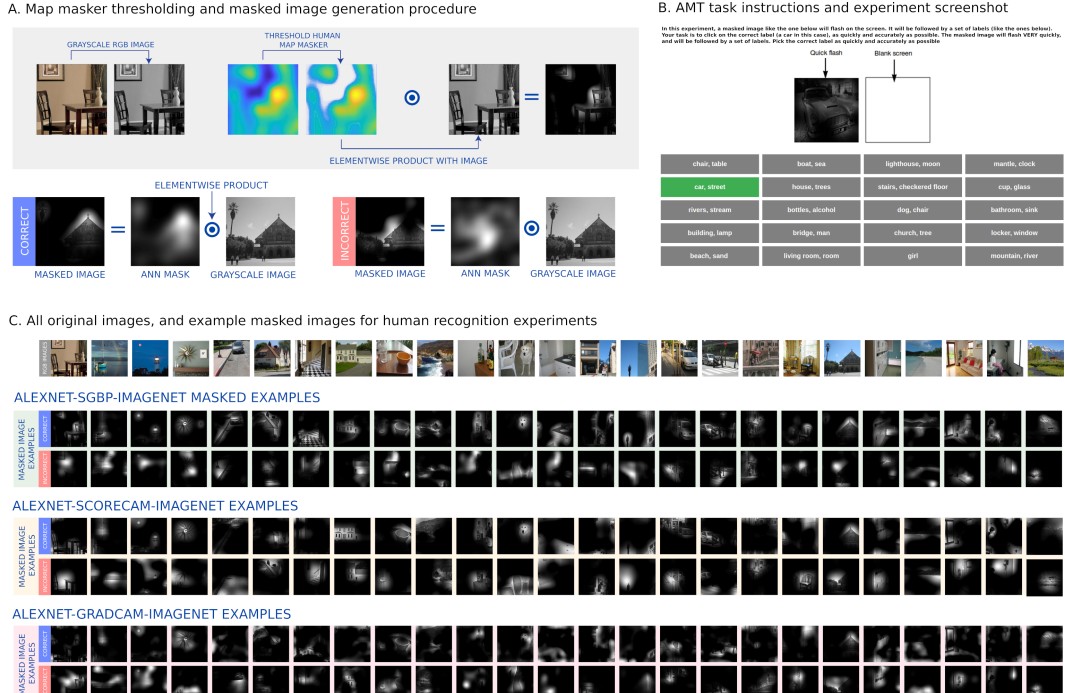

Figure S5: Human recognition experimental design and procedure. A. Masking procedure. Grayscale versions of the original images were masked by an elementwise product with smoothed ANN maps that were thresholded in order to reveal 50% of the total image area. Correct masking consisted in using the smoothed ANN map obtained for the same image as a masker, while incorrect masking consisted in using an ANN map obtained from a different image, with a random rotation applied. Only incorrect maps that were relatively weakly correlated to the correct mask (less than $r = 0.4$) were retained for incorrect masking. B. Screenshot of the Amazon Mechanical Turk (AMT) experiment instructions. C. All the original RGB images, and representative examples of correctly masked and incorrectly masked images for each experiment.

masked cases was equalized for each image. This produced a total of 2000 unique masked images (See SI Appendix Fig. S5C for representative examples, for each of the 25 images).

The masking was done as follows. The image was converted to grayscale (so as to avoid participants using color cues). The mask was an ANN map smoothed using the smoothing parameter that yielded the peak correlation to the human data. We started by thresholding the map to ensure that only 50% of its values were above 0 (as a way of roughly equalizing the amount of image area being revealed under the mask across images and conditions). We then multiplied the thresholded map and grayscale image elementwise to obtain the final masked image (see SI Appendix Fig. S5A and C).

Each participant on Amazon Mechanical Turk (AMT) completed 10 experimental trials. Their instructions were as follows: "In the task, you will see an image flash for about 200 milliseconds. Your job is simply to select the best caption (set of object words) from a list of words that will be presented to you. Choose the list of words that provide the best description of the image you saw. The images will be partly masked, and may be difficult to see. If you are unsure, just provide your best guess." See SI Appendix Fig. S5B for a screenshot of the experiment instructions from the AMT task. Participants completed 4 practice trials prior to completing the full 10 experimental trials from the image set assigned to them. Because we had several images showing similar object categories (multiple images of cars and houses), we constrained the image sets such that they never contained more than one image of a given category.

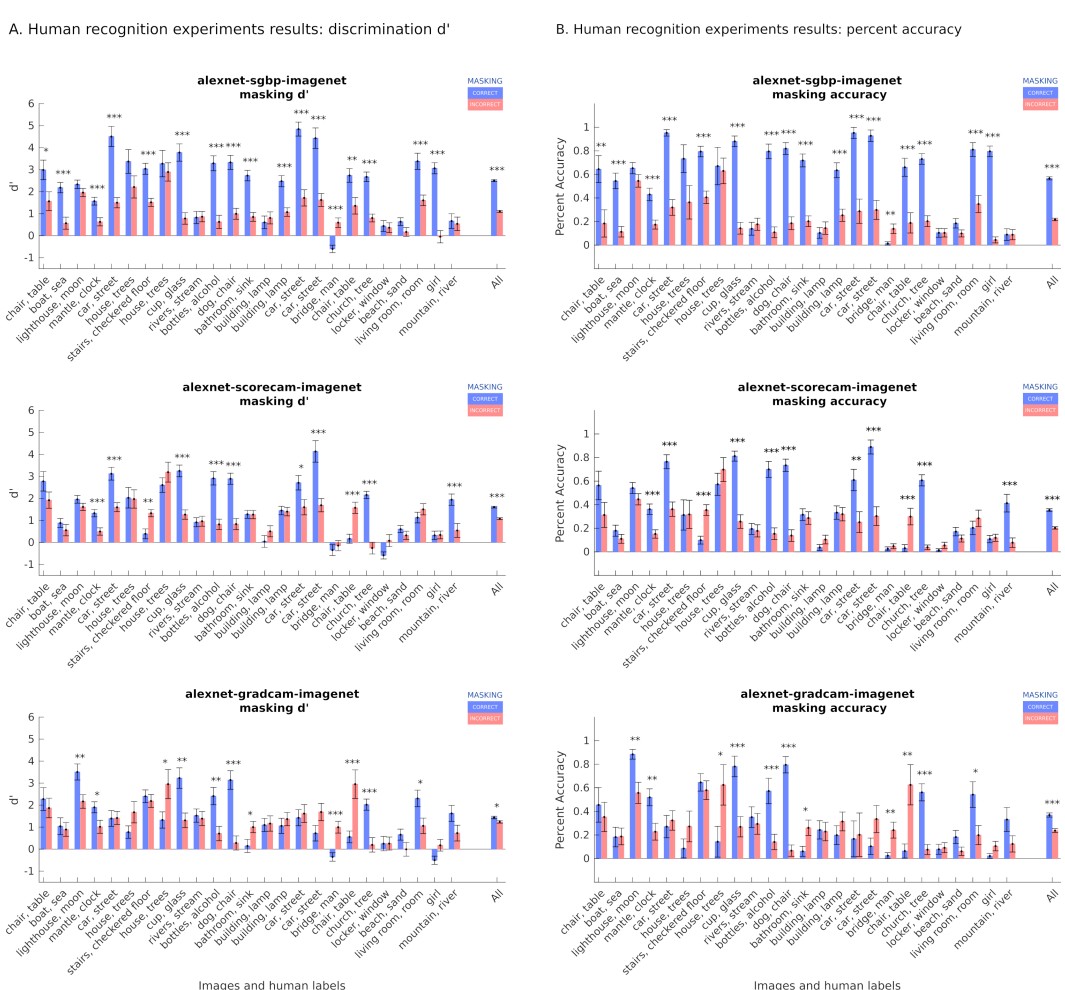

Figure S6: Human recognition experiment results. A. $d'$ results for masked images obtained using the SGBP method on AlexNet pretrained on ImageNet (first row). Also shown are $d'$ results for masked images obtained using the Score-CAM and Grad-CAM methods on AlexNet pretrained on ImageNet (second and third rows, respectively). Results show a clear main effect of correct vs. incorrect masking for most individual images ($p < 0.001$; with the Bonferroni correction for multiple comparisons applied). A significant interaction ($p < 0.001$) confirms that correctly masked images obtained using SGBP yielded higher $d'$ than correctly masked images obtained using the Score-CAM or Grad-CAM methods ($p < 0.001$; with the Bonferroni correction for multiple comparisons applied). Results are shown for participants that had a higher than 15% accuracy during the task. B. The same results are shown in terms of percent accuracy instead of $d'$.

### E.2 Crowdsourcing image captions for the human recognition experiment

We recruited a total of 160 participants but retained usable responses for 125. For each image, 5 participants were instructed to do the following "Please write three words or short phrases that summarize the contents of the image. If someone were to see these three words or phrases, they should understand the subject and context of the image." Participants were paid $0.2 to label all 25 images. To come up with the final list of labels for the human recognition experiments, we tallied the frequency of all the labels provided by all participants, and retained the top-one or top-two most frequent word labels as final labels.

## F  Human recognition experiments: analysis

### F.1  Calculating $d'$

$d'$ scores were computed for each image, and for each condition (*correct* vs. *incorrect* masking) by calculating the False Alarm (FA) rate (the number of times a given label set was selected when the image shown was not an instance of that label set, over the number of times that the presented images were not instances of that label set), and the HIT rate (the number of times that a given label set was selected when the image shown was an instance of that label set, over the number of times that all the presented images were instances of that label set). $d'$ is given by: $d' = Z(\text{HIT}) - Z(\text{FA})$ where the function $Z(p), p \in [0, 1]$, is the inverse of the cumulative distribution function of the Gaussian distribution.

### F.2  Testing the effect of masking and ANN map type

For each image and for each ANN map type (see SI Appendix Fig. S5C), we computed $d'$ using the formula described in the previous section. For each of the 25 images we computed $d'$ using HIT and FA rates from participant choices on the nAFC task. SI Appendix Fig. S6A and B show the results broken down by image, for all recognition experiments. Fig. S6A shows the results in terms of $d'$. While some images were intrinsically harder to recognize than others (with lower $d'$ scores in the correct masking conditions), the results reveal clear effects of correct vs. incorrect masking for most individual images ($p < 0.001$; we applied the Bonferroni correction for multiple comparisons). Fig. S6B shows the same results in terms of percent accuracy. In order to mitigate the problem of negative $d'$ values, we excluded data from participants that had a lower than 15% accuracy in their trials.

Paired t-tests revealed a significant difference ($p < 0.001$) between the $d'$ scores across models for the correct masking condition (blue bars in Fig. F4D of the main text and in the far right of each barplot in SI Appendix Fig. S6A). This finding confirms the prediction that human recognition accuracy is in fact significantly *more* sensitive to the visual regions revealed by one of the models with the highest peak correlations to the human PC maps, but less sensitive to those revealed by passive attention techniques on a model with a much lower peak correlation to the human PC maps, even though correct masking did produce a boost in recognition accuracy over incorrect masking for both map types ($p < 0.001$, Fig. F4D).

## G  ANN recognition experiments: design

### G.1  Masking procedure

We produced masked images using the same procedure used for the human recognition experiments as outlined in the main text (in the section called "Validation Experiment: ANN Visual Selectivity Boosts Human Recognition"), except that we used RGB (instead of grayscale) images in order to minimize distribution shift for the ANNs (See SI Appendix Fig. S7A for examples of masked RGB images). We expected a reduction in classification accuracy on masked images because of the significant distribution shift caused by masking the RGB images, regardless of the mask type. As a consequence, we simply examined the *difference* in accuracy when an image is masked with a correct mask versus with an incorrect mask. For each image, we produced one correctly masked image, and then generated 24 incorrectly masked images by pairing the image with masks of the same map type (such as using the patch ratings maps) from the other 24 images; this gave us a total of $25 \times 25$ masked images for each of the 25 images in the dataset.

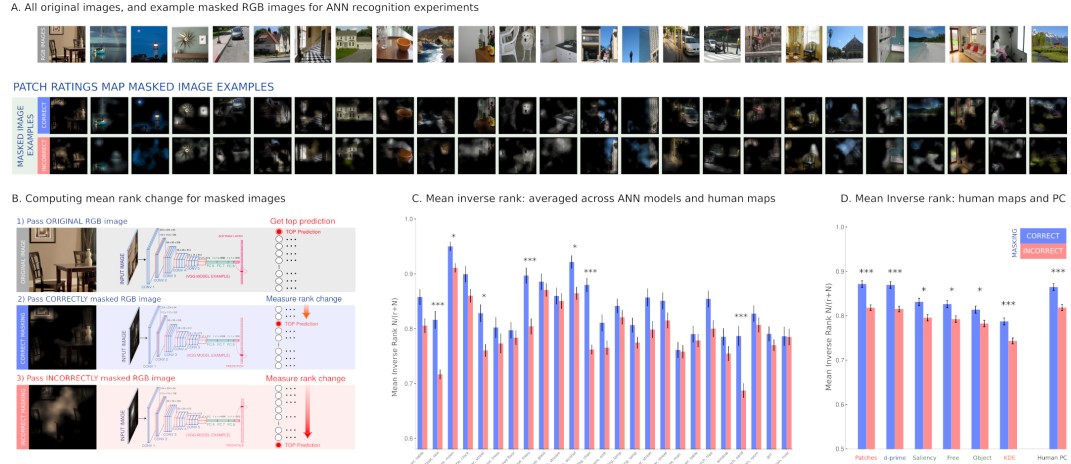

Figure S7: ANN recognition. A. All original RGB images, and masked RGB image examples. Masked image examples shown were obtained using the human patch ratings maps. B. Procedure for computing the change in rank of the top prediction. For each model, the top-1 prediction on the original unmasked image was taken as the ground truth label. We measured the rank distance (change in rank) of this prediction for the same image when correctly masked and incorrectly masked. We inverted the rank distance for comparability to the $d'$ metric used for the human experiments. We call this inverted quantity the *inverse-rank*. C. Mean inverse-rank results averaged across all ANN models and human behavioral maps. Results show clear trends for most images and significant effects ($p < 0.05$) for the lighthouse image, as well as the boat, house, dog, and beach images $p < 0.001$. We applied the Bonferroni correction for multiple comparisons. To help with comparing the ANN and human recognition results, the x-axis labels in the barplot are the labels provided by human participants, and not predictions made by the ANN models. They are ordered in the same order as the full RGB images in A. D. Mean inverse-rank result for each human behavioral map, including the Human PC factor.

## G.2 Inverse rank computation

For each model, the top-1 prediction on the unmasked image is taken as the ground truth label. The rank distance (change in rank) of this prediction due to (correct or incorrect) masking captures how a mask affects ANN recognition on the given image; for example, if the top-1 predicted label on the unmasked image attains a rank position of 5 on a masked image, the rank distance is 4. SI Appendix Fig. S7B shows a schematic of the rank distance measurement procedure. As described in the main text (Section "Validation Experiment: Human Visual Selectivity Boosts ANN Recognition" in the paper), this rank distance is inverted to align with trends in the $d'$ metric described in Appendix F.1, and we call the inverted quantity the *inverse-rank*. We computed inverse-rank as $N/(r + N)$, where $N$ is the total number of categories, and $r$ is the rank of the top-1 category predicted by the model for the full (unmasked) RGB image.

## H ANN recognition experiments: analysis

We computed the inverse-rank across all models, for all types of human maps (Section "Validation Experiment: ANN Visual Selectivity Boosts Human Recognition"), and across all 25 different masks for each of the 25 images (see SI Appendix Fig. S7C for results broken down by image). We found that the correctly masked images are more recognizable (have higher inverse-rank) than incorrectly masked images for all types of human behavioral maps (error bars represent $95\%$ confidence intervals across the 25 images). This supports the hypothesis that ANNs are sensitive to visual regions highlighted by human behavioral maps. Fig. S7D shows the results for each human behavioral map, including the Human PC map.

We performed a subsequent ANOVA analysis using the masking condition (2: correct vs. incorrect), the specific image (25), and the behavioral map type (7) as repeated measures. We find a significant

effect of masking condition (F(1,176)=184.85, $p < 0.001$) indicating that the correct masking condition gives higher mean inverse-rank. This trend is seen even across individual images (averaged across the 7 behavioral maps) in Fig. S7. We also find a significant effect of the type of behavioral map (F(6, 176)=49.65, $p < 0.001$); *i.e.,* certain types of behavioral maps produce more recognizable masks overall. For example, images masked using Human PC, patch ratings, and discrimination accuracy maps were more recognizable overall than those masked using spatial localization maps (KDEs), across both correct and incorrect masking conditions. This might be driven by the statistics of the masks and their compatibility with ANN classification. We also note that these differences are consistent with the correlational findings: Overall, the average of the peak correlations across all ANN maps were higher for the Human PC, patch ratings, and discrimination accuracy maps, than they were for the spatial localization (KDE) maps.

Finally, we examine whether different behavioral maps also capture different information, and whether this affects ANN recognizability. This is reflected in whether different behavioral maps vary in how well they distinguish between correct and incorrect masking for ANN recognition. We indeed find a significant interaction between masking condition (correct vs. incorrect) and the kind of behavioral map (F(6,176)=6.54, $p < 0.001$). This indicates that the difference in inverse-rank between correct and incorrect masking varies significantly across different behavioral maps. We find a trend that maps from behavioral measures that have higher overall peak correlation to ANN attention maps (Human PC, patch ratings, and discrimination accuracy maps) also lead to higher differences in inverse-rank across masking conditions, while measures such as eye movement patterns (that have lower overall peak correlation with ANN attention maps) give lower mean inverse-rank differences.

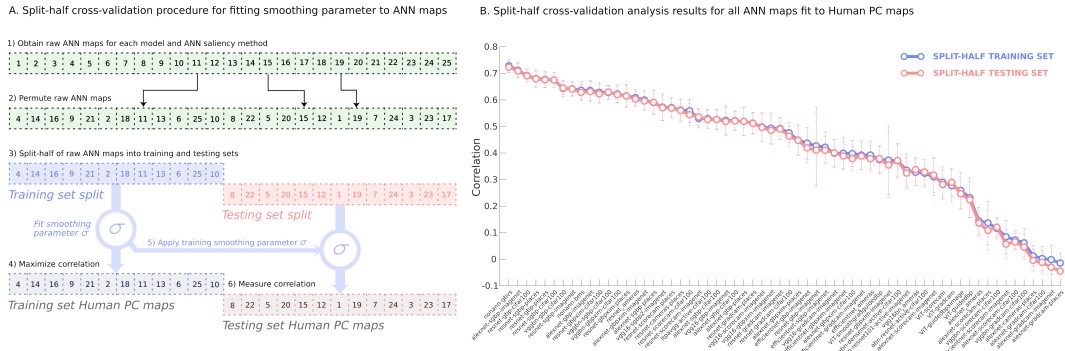

Figure S8: Split-half cross-validation analysis. A. Procedure. For each raw ANN map, we randomly permuted all the maps and divided them into a training and separate testing set. Next, we optimized a smoothing parameter $\sigma$ that maximized the average correlation of the maps in the training set to the corresponding human PC maps. We then measured the average correlation achieved between the ANN naps in the testing set and their corresponding human PC maps when smoothed using the $\sigma$ parameter fit to the training set maps. We repeated this process 100 times for each ANN map type. B. Results. The results are essentially unchanged relative to the results reported in the main text, where no cross-validation was done, and the smoothing parameter was fit to all the data. Error bars reflect 100 random permutations and splits of the ANN maps for each ANN map type. Results are ordered in terms of the peak average correlations achieved for the training set.

# I   Split-half cross-validation analysis of smoothing parameter

As a supporting analysis, we estimated peak correlations between the ANN maps and the human PC maps using split-half cross-validation when estimating the smoothing parameter (a Gaussian smoothing kernel with standard deviation $\sigma$). SI Appendix Fig. S8A illustrates the procedure. For each ANN map, we repeated the following process 100 times: We started by randomly permuting the order of the maps and splitting them into training and testing sets (the training set contained 12 maps, and the testing set contained the remaining 13 maps). Next, we fit the smoothing parameter $\sigma$ to the training set by maximizing the average correlation between the training set ANN maps and the corresponding human PC maps. We then evaluated the correlation achieved between the 13 testing set ANN maps and the corresponding human PC maps when smoothed using the same $\sigma$ parameter.

Results of the analysis are shown in SI Appendix Fig. S8B. They show a near total overlap between the peak correlations achieved for the training set maps, and those obtained for the testing set maps. In fact, the results are essentially unchanged relative to those reported in the main text: performance of smoothed test set maps using smoothing parameters $\sigma$ fit to the training set maps produced nearly identical ranges in peak correlations to the human PC (between $r = 0.73$ and $r = -0.01$ for the training set, and between $r = 0.72$ and $r = -0.04$ for the testing set). In addition, we observe a nearly identical rank order in the peak correlations to the human PC (for instance, across 100 random splits of the data, we observed an average correlation of $r = .711$ (sd $= .028$) for the peak correlation of the AlexNet SGBP maps to the human PC maps in the training set, and an average of $r = .710$ (sd $= .026$) for the peak correlation of the AlexNet SGBP maps to the human PC maps in the testing set when applying the smoothing parameter optimized to the training set). Note that we have also included non-ANN attention-based model maps in the analysis (GBVS [28] and BMS [29]), for reference.