# OpenReview forum: "Passive attention in artificial neural networks predicts human visual selectivity"
_NeurIPS.cc/2021/Conference — NeurIPS 2021 Oral_

### Official Review · Reviewer_tunC · 2021-07-12

**Rating:** 6
**Confidence:** 4

**Summary:**

In this paper, the authors systematically compare visual saliency maps obtained from deep learning models against human visual selectivity maps obtained through a series of large-scale online experiments involving a variety of perceptual tasks. The correlation between maps greatly varies depending on neural network architecture, visualization technique and behavioral task. In some cases (e.g., SGBP technique applied to AlexNet architecture) the model’s map achieves a very good match with human’s maps, suggesting a remarkable similarity between humans and machines in terms of how they select information contained in visual stimuli. This finding is further corroborated by two recognition experiments, where performance of both humans and deep nets is shown to decrease when images are masked using “incorrect” maps.

**Ethical Concerns:**

I do not see ethical issues related to the content of this paper.

**Limitations And Societal Impact:**

The authors explicitly discuss the limitations of their research approach.

**Main Review:**

Originality: This research work is interesting and novel: a systematic comparison between human visual selectivity and state-of-the-art machine vision models is still lacking. The methodology used to build human maps is taken from very recent experimental approaches and the methodology used to build ANNs maps is based on state-of-the-art techniques.

Quality: The methodology is overall sound, and the perceptual tasks used to probe human observers are appropriate. It is also remarkable that the authors implemented a wide range of passive and active attention techniques to derive saliency maps and tested a good range of deep learning architectures. However, the paper as it stands has several weaknesses, which I list here:

-	I am not fully convinced that NeurIPS would be the best venue to present these findings (maybe a more cognitively oriented conference / journal could be a better option). However, validation of neural network models against humans is one of the subject areas of NeurIPS, hence I should acknowledge that this is a matter of opinion.
-	All models considered have been trained on datasets (ImageNet and CIFAR) containing mostly single-object images, in contrast with the testing stimuli that represent complex visual scenes. I doubt that the visual saliency of a complex scene is simply the sum of visual saliency of single objects.
-	It would be useful to include a “non-deep learning” baseline to better establish whether neural networks really represent the state-of-the-art in modeling human visual selectively. I was surprised that the authors did not even mention classical saliency-based methods (for a review, see Borji and Itti, “State-of-the-art in visual attention modeling” IEEE TPAMI 2012).
-	I think the results presented in Fig. 4D are particularly informative, but only if the “Low” column refers to the AlexNet-Scorecam (as mentioned in Line 235). However, from the caption of Fig. 4 it seems that the “Low” column refers instead to the AlexNet-Gradcam maps, which I think is not particularly interesting to show given that such masking condition returns almost unperceivable images. It would be more informative to show that SGBP is still better than Scorecam, even if that method achieves a decent correlation with human PC.
-	Fig. 5 is never referred to in the text (it is mistakenly called Fig. 3 or 4).

Clarity: The paper is written in a very clear way and properly organized. I enjoyed reading it. The discussion has a good balance between technical details and more general topics related to human perception and cognitive psychology.

Significance: This work could promote further research and stimulate the interest of the deep learning community toward more systematic comparisons between neural network models and human observers.

---------------------------
I think the authors properly addressed my concerns. I am curious to read the final version of the paper with the new results. My scored changed to 6.


**Time Spent Reviewing:**

6

---

> ### Author Response · Authors · 2021-08-10
> **Response to reviewer 3 tunC**
>
> We thank the reviewer for the suggestions and comments.
>
> **Q1. Fit at NeurIPS** The CFP of NeurIPS explicitly mentions cognitive science and interdisciplinarity. Our contributions are in line with related work that has appeared in NeurIPS (e.g., Kubilius et al. (2019), Yamins et al. (2013), Harel et al. (2007)).
>
> **Q2. All models considered have been trained on datasets (ImageNet and CIFAR) containing mostly single-object images, in contrast with the testing stimuli that represent complex visual scenes. I doubt that the visual saliency of a complex scene is simply the sum of visual saliency of single objects.** We now include saliency maps obtained from AlexNet and ResNet50 models pre-trained on the MIT Places dataset (Zhou et al. (2017)), a large-scale and complex scene-recognition dataset. The results are largely consistent with our original findings. For example, ResNet-SGBP and ResNet-GBP maps for the model trained on the Places dataset had peak correlations to the human PC of r =.68 (sd =.008) and r =.677 (sd = .008), compared with the leading maps: ResNet-SGBP-ImageNet (r = .64 (sd =.008)) and ResNet-GBP-cifar100 (r =.68 (sd =.007)). Similarly, on the lower end, the peak correlations of Score-CAM maps to the human PC were small for both AlexNet trained on Places: r = .13 (sd =.008), and AlexNet trained on ImageNet  r =.32 (sd =.008).
>
> **Q3. It would be useful to include a “non-deep learning” baseline to better establish whether neural networks really represent the state-of-the-art in modeling human visual selectively.** Our focus is evaluating visual selectivity of standard ANN models in relation to human visual selectivity, so we note that we do not claim that ANN saliency represents the state-of-the-art in modeling humans. Nevertheless, we agree that including non-deep learning baseline models of visual attention (including those from the mentioned review paper) would allow useful comparisons to ANNs as models of human visual selectivity.  We therefore now cite Borji & Itti (2012), and have included two non-deep learning models in our analysis: Graph-Based Visual Saliency (GBVS) (Harel, Koch, & Perona 2007), and Boolean Map Saliency Model (BMS) (Zhang & Sclaroff 2015). These new features have the following correlations to the human PC: r =.73 (sd =.007) for the GBVS maps, and r =.63 (sd =.008) for BMS. Although the difference between the AlexNet SGBP maps and the GBVS maps in terms of their peak correlations to the Human PC was not significant (r =.71 (sd =.007) for the AlexNet SGBP maps), these results show that we can use off-the-shelf ML methods (i.e., ANN and interpretability methods as in the original submission) to produce saliency maps that can predict human visual selectivity just as well as custom models (i.e., “non-deep-learning” visual-attention models) designed specifically for that purpose.
>
> **Q4. It would be more informative to show that SGBP is still better than Score-CAM, even if that method achieves a decent correlation with the human PC.** We ran the human recognition experiment using the AlexNet Score-CAM maps as masks, and still find that percent accuracy in the correct masking condition was significantly lower than the percent accuracy measured in the correct masking condition using the AlexNet SGBP masks (Score-CAM map correct masking condition percent accuracy = 47.7% vs SGBP correct masking condition percent accuracy = 64.5%, p <.001). We have added this result to the paper. Note that we include results in terms of percent accuracy for greater interpretability of the results, based on reviewer suggestions made here.
>
> **Q5. Fig.5 is never referred to.** Thank you for noticing this. This typo, along with others we noticed since the submission, have now been corrected.
>
>
> **References:**
>
> Borji, A., & Itti, L. (2012). State-of-the-art in visual attention modeling. IEEE transactions on pattern analysis and machine intelligence, 35(1), 185-207.
>
> Harel, J., Koch, C., & Perona, P. (2007). Graph-based visual saliency.
>
> Kubilius, J., Schrimpf, M., Kar, K., Hong, H., Majaj, N. J., Rajalingham, R., ... & DiCarlo, J. J. (2019). Brain-like object recognition with high-performing shallow recurrent ANNs. arXiv preprint arXiv:1909.06161.
>
> Yamins, D., Hong, H., Cadieu, C., & DiCarlo, J. J. (2013). Hierarchical modular optimization of convolutional networks achieves representations similar to macaque IT and human ventral stream.
>
> Zhang, J., & Sclaroff, S. (2015). Exploiting surroundedness for saliency detection: a boolean map approach. IEEE transactions on pattern analysis and machine intelligence, 38(5), 889-902.
> ​​
> Zhou, B., Lapedriza, A., Khosla, A., Oliva, A., & Torralba, A. (2017). Places: A 10 million image database for scene recognition. IEEE transactions on pattern analysis and machine intelligence, 40(6), 1452-1464.

---

> > ### Comment · Reviewer_tunC · 2021-09-02
> > **Response to rebuttal**
> >
> > I think that the authors properly addressed my concerns in their revision. In light of this, I am willing to increase my score to 6 and agree with the other Reviewers for acceptance.

---

> > > ### Author Response · Authors · 2021-09-06
> > > **Response to reviewer feedback**
> > >
> > > We thank the reviewer for the response to our rebuttal. We think the suggested changes and additions made here have greatly improved the work, and they will be included in the final version of the paper.

---

### Official Review · Reviewer_Mcif · 2021-07-17

**Rating:** 7
**Confidence:** 3

**Summary:**

This paper investigates visual selectivity (i.e. which areas in images are most informative) in humans and artificial neural networks. Human attention maps are generated using different behavioral tasks, while artificial attention maps are generated from different ANN architectures. A simple ANN architecture with a passive attention mechanism is revealed as the best predictor of human visual selectivity. Using recognition experiments, the authors show that ANN-based attention maps enhance human recognition and viceversa.

**Limitations And Societal Impact:**

The authors acknowledge the main limitations of the work, as well as potential impacts.

**Main Review:**

The paper describes a comprehensive investigation of the correspondence between human and machine attention. The authors compare behavioral and eye tracking data to an array of ANN architectures with different attention mechanisms. The question of whether the areas in images that ANNs pay attention to are similar to those humans pay attention to is an interesting one, and it's great to see it addressed using a wealth of experimental data and ANN architectures.

The authors mention some previous work on this topic, but it's not very clear to me where this work sits in relation to this context (e.g. more comprehensive approach/different methods?) Furthermore, the work they cite mostly concludes that visual selectivity is different between ANNs and humans; it would be interesting to see some discussion of why the authors think their work reaches a different conclusion.

Although correlations between the different behavioral measures are reported, it would be great to also get an idea of the cross-participant reliability in the behavioral experiments. This could help understand the level of noise in the data and could even be used to estimate a noise ceiling for how well a good model would be expected to fit the data.

Furthermore, I found the combination of results from different behavioral tasks and the human PC presented in a slightly confusing way. Given that most behavioral measures do correlate fairly well, I am not sure the authors' interpretation that the human PC reflects a shared latent component is the only possible one. Would it not be possible that reducing the data in this way removes some of the variability due to noise and thus improves the results? I would also like to better understand how the human PC is calculated – is this the first component resulting from PCA and if so, how much variance does it explain? The explanation in appendix C1 seems a bit unclear on this.

Furthermore, since the human PC is used as the 'best measure' of human behavior, I am wondering why it was not used to mask the images presented to the ANN in Section 6 (as far as I can tell this section discusses separate masks based on the different behavioral tasks).

The optimization of the smoothing parameter is another aspect that could be explained a bit more clearly (especially in lines 171-175). Since the parameter was optimized based on the target correlation, I wonder whether this introduced some circularity in the analysis. Do the ANN architectures rank the same if the correlation is calculated between the raw?

It is great to see validation experiments addressing the overlap between human and machines in terms of visual selectivity. Although these experiments show that human/ANN maps improve recognition compared to randomized maps, they don't address the usefulness of these maps relative to comparable methods (e.g. masks derived from different participants or different ANN architectures). It is hard to tell from these results whether these maps really provide an 'advantage' in recognition, and it would be great to see these aspects mentioned in the discussion. Furthermore, it is difficult to judge from Figure 4D how meaningful the difference in d' is - what would be the expected d' of the 'true' model? Such difficulties in model comparison are common in this type of work, but could be discussed a bit more.

Finally, the main finding of this paper seems to be that a relatively simple model, without an active attention module, captures human visual selectivity patterns better than more complex models. However, the discussion seems relatively limited in interpreting the results, e.g. are there any conclusions that can be drawn from this about humans/ANNs? How do the authors reconcile the fact that the type of attention explains much of the variance in performance, but on average active attention models do as well as passive attention ones?

Minor comment: In places, some analyses could be described more clearly (eg. lines 195-198, the analysis done to quantify variance is described rather vaguely)

-----------------------------------------
After author response:

I thank the authors for addressing my comments so thoroughly. I think many of these issues just required more clarification, and my concerns have been addressed well in this rebuttal. In regards to Q8 - I did not mean to suggest that a different measure of accuracy should be used. By 'true' model, what I meant was that there is some individual variability in human behavior too, and so a 'great' result might actually be expected to sit below 100%. I think expanding the discussion, plus the additional checks performed by the authors, will improve the paper, and so I'm increasing my rating to 7.

**Time Spent Reviewing:**

6

---

> ### Author Response · Authors · 2021-08-10
> **Response to reviewer 2 Mcif**
>
> We thank the reviewer for the suggestions and comments.
>
> **Q1. Where does this work sit in relation to its context.** We now discuss the relevant literature, including findings that differ from our conclusions. An important departure from prior work is a more comprehensive comparison of ANN attention to multiple human behavioral measures, and the finding that guided backpropagation-based saliency methods produce more human-like saliency maps than CAM-based methods (Fig. 3A). In the same vein, we capitalized on the opportunity to compare this range of behavioral measures to the wide range of ANN models and interpretability techniques that have accumulated over the past decade. Specifically, we evaluated saliency maps produced through different combinations of models and saliency methods, and observed a wide range in the degree to which the resulting saliency maps resembled human measures. This comprehensive approach allowed us to draw different conclusions than prior work.
>
> **Q2. cross-participant reliability in the behavioral measures.** We added an analysis of participant reliability using split-half reliability. With 100 random splits of the data for each human behavioral task, we measure high reliability for all tasks. The average split-half correlations for 100 split-half pairs of the data with the Spearman-Brown correction were: ρ =.85 for the patch ratings, ρ =.88 for free fixations, ρ =.92 for cued object search fixations, ρ =.87 for saliency search fixations, ρ =.63 for the spatial memory KDEs, and ρ =.75 for the d’ maps.
>
> **Q3. I am not sure the authors' interpretation that the human PC reflects a shared latent component is the only possible one.** We agree that the PC may be more correlated because it necessarily reduces variability from each behavioral measure, however we argue that this finding was not the only possibility a-priori. We reasoned that the different behavioral maps either (a) have independent explanatory power (i.e. different components of these maps correlate with machines, with the possibility that particular behavioral maps correlate with specific ANN maps), or (b) have a shared component that correlates with the machines along with some independent noise. Our conclusion is (b) because we found that the peak correlations of the ANN maps to all of the behavioral maps were nearly always smaller than the peak correlations of the same maps to the human PC (Fig. 3B, S4). Had (a) been true, we might have expected to see the peak correlations of the ANN maps to one of the behavioral measures be significantly higher (correlations above the diagonal in the scatterplots shown in Fig 3B, S4) than the corresponding peak correlations of those same maps to the human PC. However, we observed correlations below the diagonal in nearly all cases, despite the fact that the correlations between the behavioral measures did vary somewhat (range r =.14-.86, Fig. 2B). Based on the reviewer’s concern, we have changed the discussion in the paper to reflect the different possible interpretations for our PC findings.
>
> **Q4. How is the human PC calculated? Is this the first component resulting from PCA and if so, how much variance does it explain?** Yes, we use the first component from PCA. The first PC explains 53.3% of the variance in the human behavioral measures (whereas the next three components explain 19.9%, 12.4% and 6.8% variance). The Human PC images shown in Fig. 2C, 3C, & S1E are images computed from a weighted sum of all the behavioral measures (each represented as a single concatenated image of all the individual images), where each weight in the sum is determined from the factor loading of that behavioral measure to the first PC. We have added clarifications in the SI Appendix.
>
> **Q5. I am wondering why the human PC was not used to mask the images presented to the ANN in Section 6** We have included this analysis in the paper. In line with our predictions, we find that the Mean Inverse Rank (MIR) for correctly and incorrectly masked images (using the PC maps as masks) were .85 and .80, and these were significantly different (p =.004). The difference in MIR across conditions was greater for human PC masks than it was for human masks that had lower peak correlations to the ANN maps on average, such as the cued object search fixations (MIR was .80 and .77; p =.075) for correct and incorrect masking conditions, respectively). Although this interaction did not reach significance, it is consistent with our predictions.
>
> **Q6. The optimization of the smoothing parameter is another aspect that could be explained a bit more clearly (e.g. lines 171-175). Since the parameter was optimized based on the target correlation, I wonder whether this introduced some circularity in the analysis.?** We repeated the smoothing parameter fitting using cross-validation and added it to the SI Appendix. We used 100 random splits of 50% of the data as train/test splits, and found that performance of smoothed test set maps using smoothing parameters fit to the training set maps produced nearly identical ranges in peak correlations to the human PC (between r =.73 and r =-.01 for the training set, and between r =.72 and r =-.04 for the testing set), as well as nearly identical rank order in peak correlations to the human PC (for instance, we observed an average correlation across 100 random splits of the data of r =.711 (sd =.028) for the peak correlation of the AlexNet SGBP maps to the human PC maps in the training set, and an average of r =.710 (sd =.026) for the peak correlation of the AlexNet SGBP maps to the human PC maps in the testing set when applying the smoothing parameter optimized to the training set). In fact, we found that the actual smoothing values computed from cross-validation vs. from all the data were nearly identical.
>
> **Q7. Although these experiments show that human/ANN maps improve recognition compared to randomized maps, they don't address the usefulness of these maps relative to comparable methods (e.g. masks derived from different participants or different ANN architectures).** We added a complementary analysis of percent correct: We now show that using the AlexNet SGBP maps as masks resulted in 64.4% accuracy in human performance in the correct masking condition overall, whereas using the AlexNet Grad-CAM maps in the correct masking condition resulted in a 42.7% classification accuracy. This means that the AlexNet SGBP maps do result in a significant improvement in recognition accuracy as measured in the human nAFC task.
>
> **Q8. It is difficult to judge from Fig. 4D how meaningful the difference in d' is - what would be the expected d' of the 'true' model?** We originally used d’ as it is a common measure in signal detection theory, but we now also include percent correct instead of d’. Unlike d’, percent accuracy is bounded between 0 and 100 and is easier to interpret. Considering the number of choices in the nAFC task (20 possible labels) chance performance is 5%. For the experiment using AlexNet SGBP masks, we observed 64.4% accuracy in the correct masking condition and 25.3% accuracy in the incorrect masking condition. For the experiment using AlexNet Grad-CAM masks, we observed 42.7% accuracy in the correct masking condition and 29.1% accuracy in the incorrect masking condition. These results are consistent with the d’ analysis.
>
> **Q9. The discussion seems relatively limited in interpreting the results, e.g. are there any conclusions that can be drawn from this about humans/ANNs?** Our work illustrates a new way to evaluate the psychological plausibility of ANNs, and reveals that guided backpropagation-based passive attention methods tend to produce more human-like saliency maps than CAM-based methods. In addition, we find that only certain models (such as AlexNet) probed using guided-backpropagation saliency methods show high agreement with measures of human visual selectivity. These results clearly show a lot of variability in the degree to which different models probed using different saliency methods produce visual selectivity maps that are human-like, and pave the way for evaluating what factors tend to produce the most psychologically plausible behavior. We have extended the discussion slightly to address this issue.
>
> **Q10. How do the authors reconcile the fact that the type of attention explains much of the variance in performance, but on average active attention models do as well as passive attention ones?** Although we did not observe differences between passive and active attention measures on average, a significant finding in our work is that human-inspired active-attention ANNs do not produce attention maps that are more human-like than corresponding “vanilla” (without attention modules) classification ANNs (probed using guided backpropagation-based passive attention techniques), despite the fact that such attention modules were conceived with human visual attention in mind, and with the goal of designing more human-like models. In particular, LTPA (which is essentially a VGGNet model with an added active attention module, and is trained on CIFAR-100) achieved a peak correlation with the human PC of r =.53 (sd =.007), while SGBP applied to vanilla VGG-19 (also trained on CIFAR-100) achieved a higher peak correlation of r =.63 (sd =.008), p < .001. Similarly, ABN ResNet-110 (which is a ResNet-110 model with an added active attention module) achieved a peak correlation of r =.34 (sd =.009), while SGBP applied to vanilla ResNet-110 produced a much higher peak correlation of r =.68 (sd =.006), p < .001. Finally, ABN ResNet-101 (trained on ImageNet) has a peak correlation of r =.33 (sd =.009), while SGBP applied to vanilla ResNet-101 (also trained on ImageNet) has a higher peak correlation of r =.64 (sd =.008), p < .001.

---

> ### Author Response · Authors · 2021-09-06
> **Response to reviewer feedback**
>
> We thank the reviewer for the many comments and constructive criticism. We agree that expanding the discussion to address all the issues that were raised here, as well as including all the additional checks in the final paper will greatly improve the work.

---

### Official Review · Reviewer_EJ2y · 2021-07-19

**Rating:** 9
**Confidence:** 5

**Summary:**

This paper creates artificial attention maps for 25 images, using a number of passive methods for creating attention maps crossed with four different convnet architectures, as well as two “active attention” models. These attention maps are correlated with human maps from three different experiments designed to test sensitivity or information across an image and three existing datasets of human eye tracking experiments in three different tasks using the same 25 images. The three experiments run here were 1) an experiment to measure how informative humans consider patches of an image compared to the original image, resulting in an information map; 2) a change sensitivity experiment where a dot is moved slightly, at a fixed grid over the image, resulting in a d’ map over the image; and 3) a kind of mmc with people experiment where subjects indicate where they think a dot was in an image and the next subject has to make the same judgment using the dot placed by the previous subject.

From these six experiments, they compute the first principal component of the six maps, which they term the “human PC”. They then compute the correlation of the various combinations of attention method with architecture, and find that Smooth Grad with guided backpropagation (sgbp) applied to AlexNet trained on ImageNet provides the highest correlation with the human maps. Since this is correlational, they then test how well humans can make judgments of images using the maps, and again, AlexNet with sgbp works best compared with a low-performing model, and in general, images gated by the correct maps for the image vs. the map for another image are statistically significantly better. Similarly, they came up with a measure of how well the human maps worked for the different models.


**Limitations And Societal Impact:**

yes.

**Main Review:**

In general, this review is a bit brief because the paper is obviously excellent.

Originality: This is highly original, although there have been several previous attempts to do this, that are all cited. I would have preferred that the differences between what had been done in the past and the current experiments had been described, instead of just citing the three prior works.

Quality: High. One slightly odd measurement was the way they decided how to rate the models’ performances using the human maps. They took the highest output for each network (e.g., which of the 1k outputs was highest for a network trained on imagenet) and measured how far that output fell in the ranking of outputs when given the correct human map vs. the incorrect one for the image. This was done because the masked images were out of distribution for the models. I would have preferred that they used transfer learning on the images to train a final linear layer to distinguish the 25 images, and then see which mask  changed the output the most. But I’m ok with what they did.

I would have liked it slightly better if they had completed the 2x2 table and tested humans with the human maps and the networks with the network maps, providing an upper bound to compare with.

Clarity: The paper is clearly written.

 Significance: The results are quite interesting, as they give a new methodology for rating how human-like the information used by the neural nets is, and which method for computing an attention map gives the most human-like results
.
Limitations and societal impact. They discuss how human biases might creep into their models and suggest that ways to ameliorate this should be investigated. They also mention the limitation that all this is based on only 25 images, but they justify this because of the large number of subjects needed for each experiment, which ran into the thousands.

Having read the authors' response, and the other reviews, I am now going to "make a case for this paper [I] scored so well." Since the only really negative review is now from reviewer tunC (marginally below acceptance threshold), I will respond to that specifically.
1) Cognitive Science has always had a place at NeurIPS. As a cognitive scientist myself, and a reviewer for over 30 years, I increasingly find that newer reviewers make this statement more and more often. It's a bit unfortunate that at a "neural information processing systems" conference, that reviewers ding papers that compare these systems to the best example we have of one, humans. Sigh.
2) This paper makes substantial contributions to the science! They tested over 6k subjects! They compared four architectures and k different saliency mechanisms, where k is some small integer! They backed up the correlational results with causal studies! What more do you want? Yes, they didn't test hand-designed non-deep model saliency mechanisms, and now they did. One of these mechanisms worked as well as AlexNet with sgbp. Ok, but the point here is that saliency is an emergent property of the model and mechanism used - AlexNet was not designed as a model of saliency.
3) They effectively responded to all of your complaints, as far as I can tell.





**Time Spent Reviewing:**

2

---

> ### Author Response · Authors · 2021-08-10
> **Response to reviewer 1 EJ2y**
>
> We thank the reviewer for the suggestions and comments.
>
> **Q1. I would have preferred that the differences between what had been done in the past and the current experiments had been described, instead of just citing the three prior works.** We agree that this was not clear enough in the previous version of the paper, and this issue was also raised by reviewer 2. We made our review of the relevant literature clearer and more comprehensive, and updated the intro and discussion to better situate our findings in relation to this literature. Specifically, we see our unique contributions as follows: (1) We find that ANN visual selectivity tends to correlate more with the shared component of the variability between different human behavioral measures rather than any single behavioral measure (such as overt attention in the form of eye-fixations, or d’). We conclude that ANNs are more human-like in this respect than previous work has found (Linsley et al. 2017, Linsley et al. 2018, Borji & Itti 2012). (2) By using several different ANN passive attention methods and human behavioral measures, our work introduces a new, more comprehensive way of evaluating the psychological validity of ANNs as models of human vision.
> We also believe that given how far ANNs and techniques for extracting saliency maps from them have developed, we had a unique opportunity to take stock of the wealth of models and interpretability methods now available to compare to psychophysical measures of human visual selectivity. In this spirit, we have since added results using maps extracted from a recent Visual Transformer (ViT) model to the paper, and observed that they are less similar to the human PC, with the highest performing ViT maps (SGBP) reaching a peak correlation of r = 0.38 (sd = 0.008), which was less than most other ANN maps, including the Alexnet SGBP maps (p<.001).
>
>
> **Q2. I would have liked it slightly better if they had completed the 2x2 table and tested humans with the human maps and the networks with the network maps, providing an upper bound to compare with.** We were not able to run more behavioral experiments (in order to test humans with human maps), but we did approximate the 2x2 table by computing Mean Inverse Rank (MIR) for AlexNet predictions when shown images masked with its own Smooth-Guided-Backprop (SGBP) maps (best correlation to Human PC), and its own Grad-CAM maps (worst correlation to Human PC). Although the results do not show a clear increase in MIR for images correctly masked using the SGBP maps (MIR =.876) when compared to results using images correctly masked using Grad-CAM maps (MIR =.875), the results do reveal a significant main effect of correct vs. incorrect masking in the SGBP mask experiment, and a significant interaction: a greater increase in MIR for correctly masked images relative to MIR for incorrectly masked images in the SGBP experiment (MIR =.876 vs. MIR =.8; p <.005), but a relatively smaller difference in MIR across masking conditions in the Grad-CAM experiment (MIR =.875 vs. MIR =.85 for correct and incorrect masking, respectively; p =.22). This interaction was significant (p <.001). The results of the experiment using SGBP maps as masks were comparable to the results we obtained using the Human PC images as masks (MIR =.85 and MIR =.80 for correct and incorrect masking, respectively; p <.004), although it is worth noting that the results using Human PC images was obtained by aggregating results across all the models, and not just AlexNet.
>
>
> **References**
>
> Borji, A., & Itti, L. (2012). State-of-the-art in visual attention modeling. IEEE transactions on pattern analysis and machine intelligence, 35(1), 185-207.
>
> Linsley, D., Eberhardt, S., Sharma, T., Gupta, P., & Serre, T. (2017). What are the visual features underlying human versus machine vision?. In Proceedings of the IEEE International Conference on Computer Vision Workshops (pp. 2706-2714).
>
> Linsley, D., Shiebler, D., Eberhardt, S., & Serre, T. (2018). Learning what and where to attend. arXiv preprint arXiv:1805.08819.

---

> ### Author Response · Authors · 2021-09-06
> **Response to reviewer comments and additional responses**
>
> We thank all the reviewers for their constructive feedback and comments. We have incorporated them into the new version of the paper and believe that it is much stronger as a result. These changes include a clearer discussion of how our contributions relate to prior work in both the cognitive and computer science communities, and a clearer summary of the conclusions that can be drawn from our work in both areas. We also thank the reviewers for their many other suggestions which have helped improve some of the more technical aspects of the work, including comparisons of our findings to non-deep learning models of attention, and more robust estimates of our correlational results and parameter estimation using cross-validation. All have helped to improve the quality of the work, and we are grateful for all the helpful suggestions.

---

### Decision · Program_Chairs · 2021-09-27

**Decision:**

Accept (Oral)

**Comment:**

This paper is a real tour-de-force, containing "data from 78 new experiments and 6,610 participants"  and "6 distinct behavioral tasks including visual discrimination, spatial localization, recognizability, free-viewing, cued-object search and saliency search fixations."  Though this paper is different than the majority of NeurIPS submissions, I think the breadth of the experimentation, and the relation to current trends in DL make it a great contribution and a breath of fresh air.  I think it will be of interest to NeurIPS attendees, could spark some interesting discussion and inspire future work.